# Fully Unsupervised Diversity Denoising with Convolutional Variational Autoencoders

**Mangal Prakash** *
Center for Systems Biology Dresden
Max-Planck Institute (CBG)
Dresden, Germany
prakash@mpi-cbg.de

**Alexander Krull** *†
School of Computer Science
University of Birmingham
Birmingham, UK
a.f.f.krull@bham.ac.uk

**Florian Jug**†
Center for Systems Biology Dresden
Max-Planck Institute (CBG)
Dresden, Germany
Fondazione Human Technopole, Milano, Italy
jug@mpi-cbg.de, florian.jug@fht.org

## Abstract

Deep Learning based methods have emerged as the indisputable leaders for virtually all image restoration tasks. Especially in the domain of microscopy images, various content-aware image restoration (CARE) approaches are now used to improve the interpretability of acquired data. Naturally, there are limitations to what can be restored in corrupted images, and like for all inverse problems, many potential solutions exist, and one of them must be chosen. Here, we propose DivNoising, a denoising approach based on fully convolutional variational autoencoders (VAEs), overcoming the problem of having to choose a single solution by predicting a whole distribution of denoised images. First we introduce a principled way of formulating the unsupervised denoising problem within the VAE framework by explicitly incorporating imaging noise models into the decoder. Our approach is fully unsupervised, only requiring noisy images and a suitable description of the imaging noise distribution. We show that such a noise model can either be measured, bootstrapped from noisy data, or co-learned during training. If desired, consensus predictions can be inferred from a set of DivNoising predictions, leading to competitive results with other unsupervised methods and, on occasion, even with the supervised state-of-the-art. DivNoising samples from the posterior enable a plethora of useful applications. We are $(i)$ showing denoising results for 13 datasets, $(ii)$ discussing how optical character recognition (OCR) applications can benefit from diverse predictions, and are $(iii)$ demonstrating how instance cell segmentation improves when using diverse DivNoising predictions.

## 1 Introduction

The goal of scientific image analysis is to analyze pixel-data and measure the properties of objects of interest in images. Pixel intensities are subject to undesired noise and other distortions, motivating an initial preprocessing step called *image restoration*. Image restoration is the task of removing unwanted noise and distortions, giving us clean images that are closer to the true but unknown signal.

In the past years, Deep Learning (DL) has enabled tremendous progress in image restoration (Mao et al., 2016; Zhang et al., 2017b; Zhang et al., 2017; Weigert et al., 2018). Supervised DL methods use corresponding pairs of clean and distorted images to learn a mapping between the two quality levels. The utility of this approach is especially pronounced for microscopy image data of biological samples (Weigert et al., 2017; 2018; Ouyang et al., 2018; Wang et al., 2019), where quantitative downstream analysis is essential. More recently, unsupervised content-aware image restoration

---

*Shared first authors.

†Shared last authors.

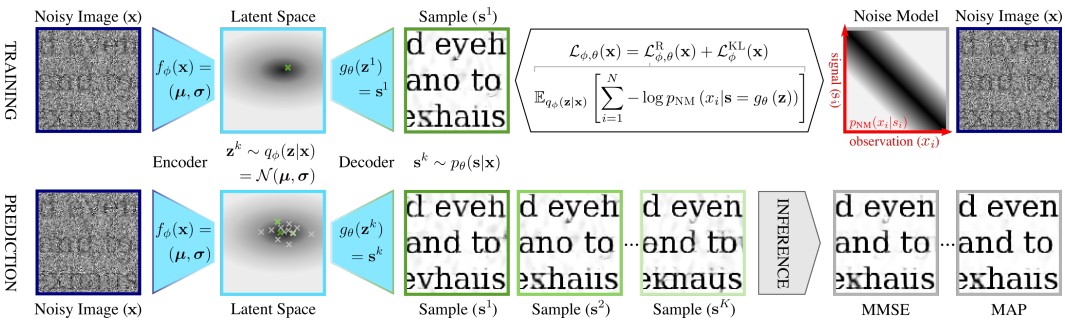

**Figure 1: Training and prediction/inference with DIVNOISING.** *(top)* A DivNoising VAE can be trained fully unsupervised, using only noisy data and a (measured, bootstrapped, or co-learned) pixel noise model $p_{NM}(x_i|s_i)$ (see main text for details). *(bottom)* After training, the encoder can be used to sample multiple $\mathbf{z}^k \sim q_\phi(\mathbf{z}|\mathbf{x})$, giving rise to diverse denoised samples $\mathbf{s}^k$. These samples can further be used to infer consensus point estimates such as a MMSE or a MAP solution.

(CARE) methods (Lehtinen et al., 2018; Krull et al., 2019; Batson & Royer, 2019; Buchholz et al., 2019) have emerged. They can, enabled by sensible assumptions about the statistics of imaging noise, learn a mapping from noisy to clean images, without ever seeing clean data during training. Some of these methods additionally include a probabilistic model of the imaging noise (Krull et al., 2020; Laine et al., 2019; Prakash et al., 2020; Khademi et al., 2020) to further improve their performance. Note that such denoisers can directly be trained on a given body of noisy images.

All existing approaches have a common flaw: distortions degrade some of the information content in images, generally making it impossible to fully recover the desired clean signal with certainty. Even an ideal method cannot know which of many possible clean images really has given rise to the degraded observation at hand. Hence, any restoration method has to make a compromise between possible solutions when predicting a restored image.

Generative models, such as VAEs, are a canonical choice when a distribution over a set of variables needs to be learned. Still, so far VAEs have been overlooked as a method to solve unsupervised image denoising problems. This might also be due to the fact that vanilla VAEs (Kingma & Welling, 2014; Rezende et al., 2014) show sub-par performance on denoising problems (see Section 6).

Here we introduce DIVNOISING, a principled approach to incorporate explicit models of the imaging noise distribution in the decoder of a VAE. Such noise models can be either measured or derived (bootstrapped) from the noisy image data alone (Krull et al., 2020; Prakash et al., 2020). Additionally we propose a way to co-learn a suitable noise model during training, rendering DIVNOISING fully unsupervised. We show on 13 datasets that fully convolutional VAEs, trained with our proposed DIVNOISING framework, yield competitive results, in 8 cases actually becoming the new state-of-the-art (see Fig. 2 and Table 1). Still, the key benefit of DIVNOISING is that the method does not need to commit to a single prediction, but is instead capable of generating diverse samples from an approximate posterior of possible true signals. (Note that point estimates can still be inferred if desired, as shown in Fig. 4.) Other unsupervised denoising methods only provide a single solution (point estimate) of that posterior (Krull et al., 2019; Lehtinen et al., 2018; Batson & Royer, 2019) or predict an independent posterior distribution of intensities per pixel (Krull et al., 2020; Laine et al., 2019; Prakash et al., 2020; Khademi et al., 2020). Hence, DIVNOISING is the first method that learns to approximate the posterior over meaningful structures in a given body of images.

We believe that DIVNOISING will be hugely beneficial for computational biology applications in biomedical imaging, where noise is typically unavoidable and huge datasets need to be processed on a daily basis. Here, DIVNOISING enables unsupervised diverse SOTA denoising while requiring only comparatively little computational resources, rendering our approach particularly practical.

Finally, we discuss the utility of diverse denoising results for OCR and showcase it for a ubiquitous analysis task in biology – the instance segmentation of cells in microscopy images (see Fig. 5). Hence, DIVNOISING has the potential to be useful for many real-world applications and will not only generate state-of-the-art (SOTA) restored images, but also enrich quantitative downstream processing.

## 2  RELATED WORK

**Classical Denoising.**  The denoising problem has been addressed by a variety of filtering approaches. Arguably some of the most prominent ones are Non-Local Means (Buades et al., 2005) and BM3D (Dabov et al., 2007), which implement a sophisticated non-local filtering scheme. A comprehensive survey and in-depth discussion of such methods can be found in (Milanfar, 2012).

**DL Based Denoising.** Deep Learning methods which directly learn a mapping from a noisy image to its clean counterpart (see *e.g.* (Zhang et al., 2017a) and (Weigert et al., 2018)) have outperformed classical denoising methods in recent years. Two well known contributions are the seminal works by Zhang et al. (2017a) and later by Weigert et al. (2018). More recently, a number of unsupervised variations have been proposed, and in Section 1 we have described their advantages and disadvantages in detail. One additional interesting contribution was made by Ulyanov et al. (2018), introducing a quite different kind of unsupervised restoration approach. Their method, *Deep Image Prior*, trains a network separately for each noisy input image in the training set, making this approach computationally rather expensive. Furthermore, training has to be stopped after a suitable but a priori unknown number of training steps.

Recently, Quan et al. (2020) proposed an interesting method called SELF2SELF which trains a U-NET like architecture requiring only single noisy images. The key idea of this approach is to use blind spot masking, similar to Krull et al. (2019), together with *dropout* (Srivastava et al., 2014), which avoids overfitting and allows sampling of diverse solutions. Similar to DIVNOISING, the single denoised result is obtained by averaging many diverse predictions. Diverse results obtained via dropout are generally considered to capture the so called *epistemic* or *model  uncertainty* (Gal & Ghahramani, 2016; Lakshminarayanan et al., 2017), *i.e.* the uncertainty arising from the fact that we have a limited amount of training data available. In contrast, DIVNOISING combines a VAE and a model of the imaging noise to capture what is known as *aleatoric* or *data uncertainty* (Böhm et al., 2019; Sensoy et al., 2020), *i.e.* the unavoidable uncertainty about the true signal resulting from noisy measurements. Like in Ulyanov et al. (2018), also SELF2SELF trains separately on each image that has to be denoised. While this renders the method universally applicable, it is computationally prohibitive when applied to large datasets. The same is true for real time applications such as facial denoising. DIVNOISING, on the other hand, is trained only once on a given body of data. Afterwards, it can be efficiently applied to new images. A detailed comparison of SELF2SELF and DIVNOISING in terms of denoising performance, run time and GPU memory requirements can be found in Appendix A.14 and Appendix Table 2.

**Denoising (Variational) Autoencoders.** Despite the suggestive name, *denoising variational autoencoders* (Im et al., 2017) are not solving denoising problems. Instead, this method proposes to add noise to the input data in order to boost the quality of encoder distributions. This, in turn, can lead to stronger generative models. Other methods also follow a similar approach to improve overall performance of autoencoders (Vincent et al., 2008; 2010; Jiao et al., 2020).

**VAEs for Diverse Solution Sampling.** Although not explored in the context of unsupervised denoising, VAEs are designed to sample diverse solutions from trained posteriors. The probabilistic U-NET (Kohl et al., 2018; 2019) uses conditional VAEs to learn a conditional distribution over segmentations. Baumgartner et al. (2019) improve the diversity of segmentation samples by introducing a hierarchy of latent variables to model segmentations at multiple resolutions. Unlike DIVNOISING, both methods rely on paired training data. Nazabal et al. (2020) employ VAEs to learn the distribution of incomplete and heterogeneous data in a fully unsupervised manner. Babaeizadeh et al. (2017) build upon a VAE style framework to predict multiple plausible future frames of videos conditioned on given context frames. A variational inference approach was used by Balakrishnan et al. (2019) to generate multiple deprojected samples for images and videos collapsed in either spatial or temporal dimensions. Unlike all these approaches, we address the uncertainty introduced by common imaging noise and show how denoised samples can improve downstream processing.

## 3  THE DENOISING TASK

Image restoration is the task of estimating a clean signal $\mathbf{s} = (s_1, \ldots, s_N)$ from a corrupted observation $\mathbf{x} = (x_1, \ldots, x_N)$, where $s_i$ and $x_i$, refer to the respective pixel intensities. The corrupted $\mathbf{x}$ is thought to be drawn from a probability distribution $p_{\mathrm{NM}}(\mathbf{x}|\mathbf{s})$, which we call the *observation*

*likelihood* or the *noise model*. In this work we focus on restoring images that suffer from insufficient illumination and detector/camera imperfections. Contrary to existing methods, DIVNOISING is designed to capture the inherent uncertainty of the denoising problem by learning a suitable posterior distribution. Formally, the posterior we are interested in is $p(\mathbf{s}|\mathbf{x}) \propto p(\mathbf{x}|\mathbf{s})p(\mathbf{s})$ and depends on two components: the *prior* distribution $p(\mathbf{s})$ of the signal as well as the observation likelihood $p_{\mathrm{NM}}(\mathbf{x}|\mathbf{s})$ we introduced above. While the prior is a highly complex distribution, the likelihood $p(\mathbf{x}|\mathbf{s})$ of a given imaging system (camera/microscope) can be described analytically (Krull et al., 2020).

**Models of Imaging Noise.** The noise model is usually thought to factorize as a product of pixels, implying that the corruption, given the underlying signal, is occurring independently in each pixel as

$$p(\mathbf{x}|\mathbf{s}) = \prod_i^N p_{\mathrm{NM}}(x_i|s_i). \tag{1}$$

This assumption is known to hold true for Poisson shot noise and camera readout noise (Zhang et al., 2019; Krull et al., 2020; Prakash et al., 2020). We will refer to the probability $p_{\mathrm{NM}}(x_i|s_i)$ of observing a particular noisy value $x_i$ at a pixel $i$ given clean signal $s_i$ as the *pixel noise model*. Various types of pixel noise models have been proposed, ranging from physics based analytical models (Zhang et al., 2019; Luisier et al., 2010; Foi et al., 2008) to simple histograms (Krull et al., 2020). In this work, we follow the Gaussian Mixture Model (GMM) based noise model description of (Prakash et al., 2020). The parameters of a noise model can be estimated whenever pairs $(\mathbf{x}', \mathbf{s}')$ of corresponding noisy and clean calibration images are available (Krull et al., 2020). The signal $\mathbf{s}' \approx \frac{1}{M} \sum_{j=0}^{M} \mathbf{x}'^j$ can then be computed by averaging these noisy observations (Prakash et al., 2020). In a case where no calibration data can be acquired, $\mathbf{s}'$ can be estimated by a bootstrapping approach (Prakash et al., 2020). Later, we additionally show how a suitable noise model can be co-learned during training.

## 4 THE VARIATIONAL AUTOENCODER (VAE)

We want to briefly introduce the VAE approach introduced by Kingma & Welling (2014). A more complete introduction to the topic can be found in (Doersch, 2016; Kingma & Welling, 2019). VAEs are generative models, capable of learning complex distributions over images $\mathbf{x}$, such as hand written digits (Kingma & Welling, 2014) or faces (Huang et al., 2018). To achieve this, VAEs use a latent variable $\mathbf{z}$ with a fixed (usually a unit normal distribution) prior $p(\mathbf{z})$ and describe

$$p_\theta(\mathbf{x}) = \int p_\theta(\mathbf{x}|\mathbf{z})p(\mathbf{z})d\mathbf{z}. \tag{2}$$

Like conventional autoencoders, they consist of two components: A *decoder* network $g_\theta(\mathbf{z})$ that takes a point in latent space and maps it to a distribution $p_\theta(\mathbf{x}|\mathbf{z})$ in image space and an *encoder* network $f_\phi(\mathbf{x})$, which takes an observed image and maps it to a distribution $q_\phi(\mathbf{z}|\mathbf{x})$ in latent space. By $\phi$ and $\theta$, we denote network parameters of the encoder and decoder, respectively.

Note that the decoder alone (together with a suitable prior $p(\mathbf{z})$) is sufficient to completely describe the generative model in Eq. 2. It is usually modelled to factorize over pixels

$$p_\theta(\mathbf{x}|\mathbf{z}) = \prod_{i=1}^N p_\theta(x_i|\mathbf{z}), \tag{3}$$

where $p_\theta(x_i|\mathbf{z})$ is a normal distribution, with its mean and variance predicted by the decoder network network $g_\theta(\mathbf{x})$. The encoder distribution is modelled in a similar fashion, factorizing over the dimensions of the latent space.

Training for VAEs consists of adjusting the parameters $\theta$ to make sure that Eq. 2 fits the distribution of training images $\mathbf{x}$. Kingma *et al.* show that this can be achieved with the help of the encoder by jointly optimizing $\phi$ and $\theta$ to minimize the loss $\mathcal{L}_{\phi,\theta}(\mathbf{x}) = \mathcal{L}_{\phi,\theta}^{\mathrm{R}}(\mathbf{x}) + \mathcal{L}_\phi^{\mathrm{KL}}(\mathbf{x})$, where $\mathcal{L}_{\phi,\theta}^{\mathrm{R}}(\mathbf{x}) = \mathbb{E}_{q_\phi(\mathbf{z}|\mathbf{x})}[-\log p_\theta(\mathbf{x}|\mathbf{z})] = \mathbb{E}_{q_\phi(\mathbf{z}|\mathbf{x})}\left[\sum_{i=1}^N -\log p_\theta(x_i|\mathbf{z})\right]$, and $\mathcal{L}_\phi^{\mathrm{KL}}(\mathbf{x})$ is the KL divergence $\mathbb{KL}(q_\phi(\mathbf{z}|\mathbf{x})||p(\mathbf{z}))$. While $\mathcal{L}_\phi^{\mathrm{KL}}(\mathbf{x})$ can be computed analytically, the expected value in $\mathcal{L}_{\phi,\theta}^{\mathrm{R}}(\mathbf{x})$ is approximated by drawing a single sample $\mathbf{z}^1$ from $q_\phi(\mathbf{z}|\mathbf{x})$ and using the *reparametrization trick* by Kingma & Welling (2014) for gradient computation.

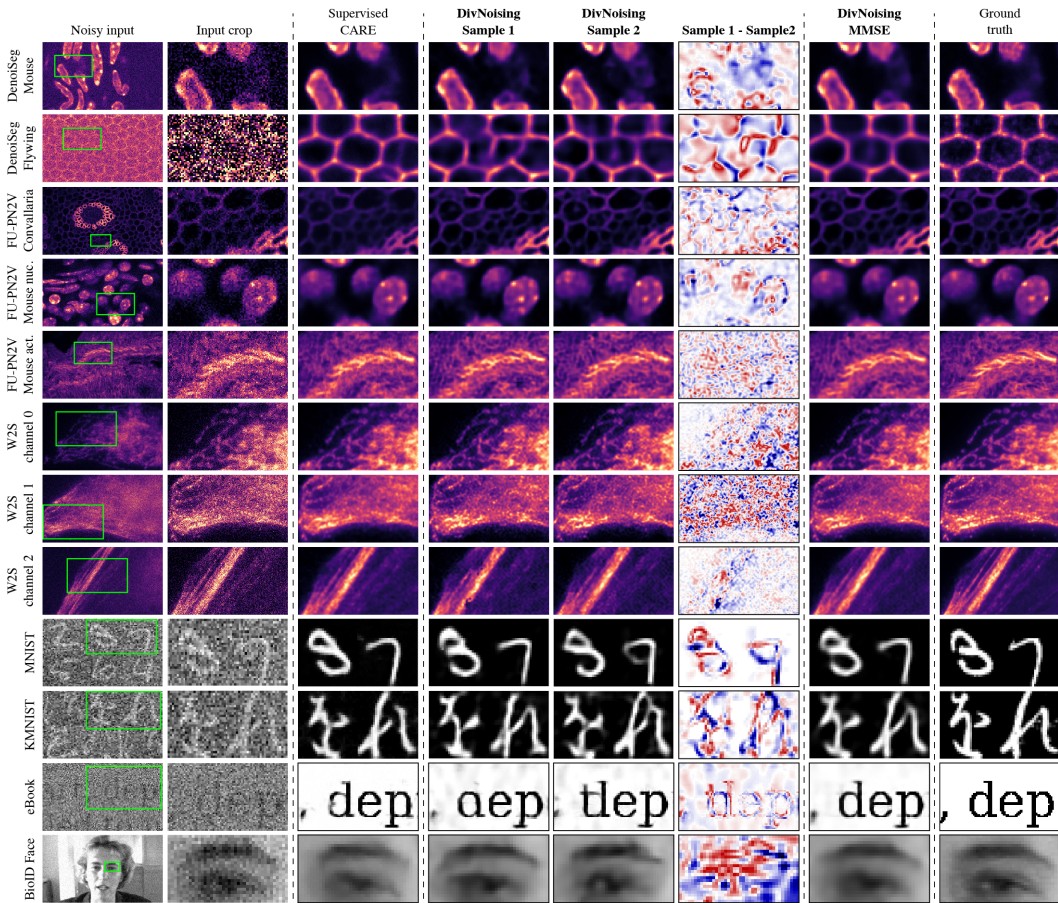

Figure 2: **Qualitative denoising results.** We compare two DIVNOISING samples, the MMSE estimate (derived by averaging 1000 sampled images), and results by the supervised CARE baseline. The diversity between individual samples is visualized in the column of difference images. (See Appendix A.9 for additional images of DIVNOISING results.)

## 5  DIVNOISING

In DIVNOISING, we build on the VAE setup but interpret it from a denoising-specific perspective. We assume that images have been created from a clean signal $\mathbf{s}$ via a known noise model, i.e., $\mathbf{x} \sim p_{\mathrm{NM}}(\mathbf{x}|\mathbf{s})$. To account for this within the VAE setup, we replace the generic normal distribution over pixel intensities in Eq. 3 with a known noise model $p_{\mathrm{NM}}(\mathbf{x}|\mathbf{s})$ (see Eq. 1). We get $p_\theta(\mathbf{x}|\mathbf{z}) = p_{\mathrm{NM}}(\mathbf{x}|\mathbf{s}) = \prod_i^N p_{\mathrm{NM}}(x_i|s_i)$, with the decoder now predicting the signal $g_\theta(\mathbf{z}) = \mathbf{s}$. Together with $p(\mathbf{z})$ and the noise model, the decoder now describes a full joint model for all three variables, including the signal:

$$p_\theta(\mathbf{z}, \mathbf{x}, \mathbf{s}) = p_{\mathrm{NM}}(\mathbf{x}|\mathbf{s})p_\theta(\mathbf{s}|\mathbf{z})p(\mathbf{z}), \qquad (4)$$

where we assume that $p_{\mathrm{NM}}(\mathbf{x}|\mathbf{s}, \mathbf{z}) = p_{\mathrm{NM}}(\mathbf{x}|\mathbf{s})$. For a given $\mathbf{z}^k$, as for standard VAEs, the decoder describes a distribution over noisy images $p(\mathbf{x}|\mathbf{z})$. The corresponding clean signal $\mathbf{s}^k$, in contrast, is deterministically defined. Hence, $p_\theta(\mathbf{s}|\mathbf{z})$ is a Dirac distribution centered at $g_\theta(\mathbf{z})$.

**Training.**   Considering Eq. 1, the reconstruction loss becomes $\mathcal{L}^{\mathrm{R}}_{\phi,\theta}(\mathbf{x}) = \mathbb{E}_{q_\phi(\mathbf{z}|\mathbf{x})}\left[\sum_{i=1}^N -\log p(x_i|\mathbf{s} = g_\theta(\mathbf{z}))\right]$. Apart from this modification, we can follow the standard VAE training procedure, just as described in Section 4. Since we have only modified how the decoder distribution is modeled, we can assume that the training procedure still produces $(i)$ a model describing the distribution of our training data, while $(ii)$ making sure that the encoder distribution well approximates the distribution of the latent variable given the image. A complete derivation of the DIVNOISING loss (from probability model perspective) can be found in Appendix A.12.

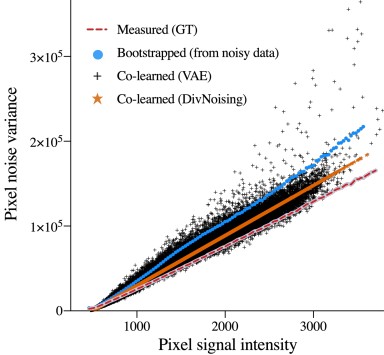

Figure 3: **Sensibility of Noise Models.** For each predicted signal intensity (x-axis), we show the variance of noisy observations (y-axis). The plot is generated from experiments on the *Convallaria* dataset. The dashed red line shows the true noise distribution (measured from pairs of noisy and clean calibration data). This true distribution, as well as the noise model created via bootstrapping, and the noise model we co-learned with DIVNOISING, show simple (approximately) linear relationships between signal intensities and noise variance. Such a relationship is known to coincide with the physical reality of Poisson noise (shot noise) (Zhang et al., 2019). The implicitly learned noise model of the vanilla VAE has to independently predict the noise variance for each pixel. Its predictions clearly deviate from the true linear relationship. See Appendix A.13 for results on *BioID Face* dataset and more details.

| | Dataset | Fully Unsupervised | | | Unsup. ($p_{NM}$ requ.) | | Supervised |
|---|---|---|---|---|---|---|---|
| | | N2V | Vanilla VAE | DivNoising | PN2V | DivNoising | CARE |
| FU-PN2V | Convallaria | 35.73±0.037 | 36.57±0.033 | *36.78±0.007* | 36.47±0.031 | **36.90±0.004** | 36.71±0.026 |
| | ↳ Bootstrapped | | | | 36.70±0.012 | 36.64±0.023 | |
| | Mouse Act. | 33.39±0.014 | 33.46±0.158 | *33.82±0.006* | 33.86±0.018 | **33.99±0.004** | 34.20±0.021 |
| | Mouse Nuc. | 35.84±0.015 | 35.84±0.023 | *36.05±0.052* | **36.35±0.018** | 36.26±0.047 | 36.58±0.019 |
| W2S | Ch.0 (avg1) | ***34.59±0.041*** | 33.02±0.147 | 34.24±0.006 | - | 34.13±0.002 | 35.22±0.069 |
| | Ch.1 (avg1) | 32.11±0.030 | 31.36±0.041 | ***32.22±0.021*** | - | **32.22±0.013** | 32.88±0.021 |
| | Ch.2 (avg1) | 35.04±0.073 | 33.72±0.187 | ***35.24±0.028*** | 32.79±0.085 | 35.18±0.020 | 35.91±0.030 |
| | Ch.0 (avg16) | 39.01±0.019 | 39.27±0.192 | *39.45±0.036* | 39.36±0.103 | **39.63±0.007** | 42.35±0.012 |
| | Ch.1 (avg16) | 37.91±0.059 | 38.33±0.021 | *38.41±0.018* | **38.46±0.012** | 38.39±0.007 | 39.64±0.061 |
| | Ch.2 (avg16) | 40.30±0.023 | 40.24±0.043 | ***40.56±0.019*** | 40.36±0.091 | 40.41±0.041 | 42.03±0.027 |
| DenoiSeg | Mouse | 33.84±0.070 | *34.06±0.003* | *34.06±0.005* | **34.19±0.037** | 34.13±0.003 | 35.11±0.016 |
| | Flywing | 24.79±0.034 | 24.88±0.045 | *24.92±0.016* | 24.85±0.036 | **25.02±0.024** | 25.79±0.014 |
| | Mouse s&p | 32.98±0.020 | 23.62±0.084 | *35.19±0.030* | 29.67±0.079 | **36.21±0.015** | 37.03±0.016 |
| | BioID Face | 32.34±0.080 | 32.58±0.022 | *33.02±0.020* | **33.76±0.079** | 33.12±0.039 | 35.06±0.051 |

Table 1: **Quantitative results.** For all experiments, we compare all results in terms of mean Peak Signal-to-Noise Ratio (PSNR in dB) and ±1 standard error over 5 runs. Overall best performance indicated by being underlined, best unsupervised method in bold, and best fully unsupervised method in italic. For many datasets, DIVNOISING is the unsupervised SOTA, typically not being far behind the supervised CARE results.

**Prediction.** While we can use the trained VAE to generate images from $p_\theta(\mathbf{x})$ (see Appendix A.5), here we are mainly interested in denoising. Hence, we desire access to the posterior $p(\mathbf{s}|\mathbf{x})$, *i.e.* the distribution of possible clean signals $\mathbf{s}$ given a noisy observation $\mathbf{x}$. Assuming the encoder and decoder are sufficiently well trained, samples $\mathbf{s}^k$ from an approximate posterior can be obtained by $(i)$ feeding the noisy image $\mathbf{x}$ into our encoder, $(ii)$ drawing samples $\mathbf{z}^k \sim q_\phi(\mathbf{z}|\mathbf{x})$, and $(iii)$ decoding the samples via the decoder to get $\mathbf{s}^k = g_\theta(\mathbf{z}^k)$.

**Inference.** Given a set of posterior samples $\mathbf{s}^k$ for a noisy image $\mathbf{x}$, we can infer different consensus estimates (point estimates). We can, for example, approximate the MMSE estimate (see Fig. 2), by averaging many samples $\mathbf{s}^k$. Alternatively, we can attempt to find the *maximum a posteriori* (MAP) estimate, *i.e.* the most likely signal given the noisy observation $\mathbf{x}$, by finding the mode of the posterior distribution. For this purpose, we iteratively use the mean shift algorithm (Cheng, 1995) with decreasing bandwidth to find the mode of our sample set (see Fig. 4 and Appendix A.4).

**Fully Unsupervised DivNoising.** So far we explained our setup under the assumption that the noise model can either be measured with paired calibration images, or bootstrapped from noisy data (Prakash et al., 2020). Here, we propose yet another alternative approach of co-learning the noise model directly from noisy data during training. More concretely, this is enabled by a simple modification to the DIVNOISING decoder. We assume that the noise at each pixel $i$ follows a normal distribution with its variance being a linear function of $\mathbf{s}_i$, *i.e.*, $\sigma_i^2 = a\mathbf{s}_i + b$. Linearity is motivated by noise properties in low-light settings (Faraji & MacLean, 2006; Jezierska et al., 2011). The learnable network parameters $a$ and $b$ are co-optimized during training. Since variances cannot be negative, we additionally constrain the predicted values for $\sigma_i^2$ to be positive (see Appendix A.3 for details).

**Denoising with Vanilla VAEs.** While not originally intended for denoising tasks, we were curious to see how vanilla VAEs perform when applied to these problems. Just like fully unsupervised DIVNOISING, also the vanilla VAE does not require a noise model. It does, instead, directly predict per-pixel mean and variance (see Section 4), leaving the possibility open that the right values could be learned. However, here the decoder is not restricted to make each pixel's variance a function of predicted signal. We investigate the denoising performance of the vanilla VAE in Section 6 and show in Fig. 5 that the predicted variances significantly diverge from ground truth noise distributions.

**Signal Prior in DIVNOISING.** Classical denoising methods often explicitly model the image/signal prior $p(\mathbf{s})$ *e.g.* as smoothness priors (Grimson & Grimson, 1981; Li, 1994), non-local similarity priors (Buades et al., 2005; Dabov et al., 2007), sparseness priors (Tibshirani, 1996) etc., assuming specific properties of the images at hand. They effectively assign the same probability to all images/signals sharing *e.g.* the same level of smoothness. However, the true distribution $p(\mathbf{s})$ of clean signals (*e.g.* for a particular experimental setup in a fluorescence microscope) is generally more complex. Instead of explicitly modelling $p(\mathbf{s})$, DIVNOISING only implicitly describes $p_\theta(\mathbf{s}) = \int p_\theta(\mathbf{s}|\mathbf{z})p(\mathbf{z})d\mathbf{z}$ as integral over all possible values of $\mathbf{z}$. We recall that the prior $p(\mathbf{z})$ is assumed to be the unit Gaussian distribution and the conditional distribution $p_\theta(\mathbf{s}|\mathbf{z})$ is learned by the decoder network as the Dirac distribution centered at $g_\theta(\mathbf{z})$. Depending on its parameters $\theta$, the network will implement the function differently, leading to a different $p_\theta(\mathbf{s}|\mathbf{z})$, and ultimately to a different $p_\theta(\mathbf{s})$. This implicit distribution is quite powerful and can capture complex structures. See Appendix A.5 for samples obtained from this signal prior for different datasets.

## 6 DATA, EXPERIMENTS, RESULTS

We quantitatively evaluated the performance of DIVNOISING on 13 publicly available datasets (see Appendices A.1 and A.2 for data details), 9 of which are subject to high levels of intrinsic (real world) noise. To 4 others we synthetically added noise, hence giving us full knowledge about the nature of the added noise.

**Denoising Baselines.** We choose state-of-the-art baseline methods to compare against DIVNOISING, namely, the supervised CARE (Weigert et al., 2018) and the unsupervised methods NOISE2VOID (N2V) (Krull et al., 2019) and Probabilistic NOISE2VOID (PN2V) (Krull et al., 2019). All baselines use the available implementations of (Krull et al., 2020) and, as long as not specified otherwise, make use of a depth 3 U-NET with 1 input channel and 64 channels in the first layer. As an additional baseline, we choose vanilla VAEs with the same network architecture as DIVNOISING, but predicting per pixel mean and variance independently. Training is performed using the ADAM (Kingma & Ba, 2015) optimizer for 200 epochs with 10 steps per epoch with a batch size of 4 and a virtual batch size of 20 for N2V and CARE and a batch size of 1 and a virtual batch size of 20 for PN2V, an initial learning rate of 0.001, and the same basic learning rate scheduler as in (Krull et al., 2020). All baselines use on the fly data augmentation (flipping and rotation) during training.

**Training Details.** In all experiments we use rather small, fully convolutional VAE networks, with either 200k or 713k parameters (see Appendix A.3). For all experiments on intrinsically noisy microscopy data, validation and test set splits follow the ones described in the respective publication. In contrast to the synthetically noisy data, no apriori noise model is known for microscopy datasets. For these datasets, we used GMM-based noise models (Prakash et al., 2020; Khademi et al., 2020), which are measured from calibration images, as well as co-learned noise models. For the *W2S* datasets, no dedicated calibration samples to create noise models are available. Hence, for this dataset, we use the available clean ground truth images and all noisy observations of the training data to learn a GMM-based noise model. All GMM noise models use 3 Gaussians and 2 coefficients each. Find more training details in Appendix A.3.

**Denoising Results.** In Table 1, we report denoising performance of all experiments we conducted in terms of peak signal-to-noise ratio (PSNR) with respect to available ground truth images. The DIVNOISING results (using the MMSE estimate from 1000 averaged samples) are typically either on par or even beyond the denoising quality reached by the baselines in the 'fully unsupervised' category, as well as the 'unsupervised with noise model' category.

Note that sampling is very efficient. For all presented experiments sampling 1000 images consistently took less than 7 seconds (see Table 3 in Appendix A.15 for precise sampling times). The effect of averaging a different number of samples is explored in Appendix A.8.

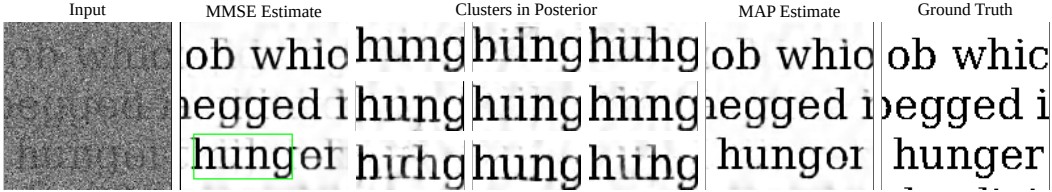

**Figure 4: Exploring the learned posterior.** The MMSE estimate (average of 10k samples) shows faintly overlaid letters as a consequence of ambiguities in noisy input. Among these samples from the posterior, we use mean shift clustering (on smaller crops) to identify diverse and likely points in the posterior. We show 9 such cluster centers in no particular order. We also obtain an approximate MAP estimate (see Supplementary Material), which has most artifacts of the MMSE solution removed.

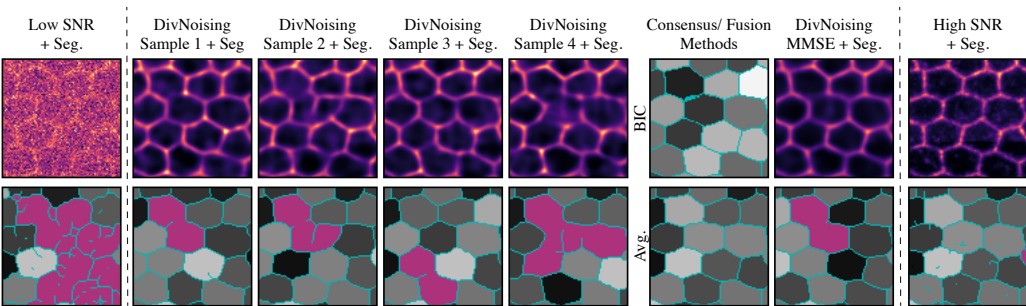

**Figure 5: DIVNOISING enables downstream segmentation.** We show input images (upper row) and results of a fixed (untrained) segmentation pipeline (lower row). Cells that were segmented incorrectly (merged or split) are indicated in magenta. While segmentations of the noisy raw data are of very poor quality, sampled DIVNOISING results give rise to much better and diverse solutions (cols. 2-5). We then use two label fusion methods to find consensus segmentations (col. 6), which are even outperforming segmentation results on high SNR (GT) images. Quantitative results are presented in Appendix Fig. 11.

DIVNOISING MMSE is typically, as expected, slightly behind the performance of the fully supervised baseline CARE (Weigert et al., 2018). Additionally, on *FU-PN2V Convallaria* we have demonstrated that a suitable noise model for DIVNOISING can be created via bootstrapping (Prakash et al., 2020; Khademi et al., 2020). We also compare against Deep Image Prior (DIP) on *DenoiSeg Flywing* dataset as it has smallest number of test images and DIP has to be trained for each image. DIP achieves PSNR of $24.67 \pm 0.050$dB compared to $25.02 \pm 0.024$dB with DIVNOISING.

Due to the extensive computational requirements of SELF2SELF, we cannot run the method on all images in any of our dataset. Instead, we run it on single, randomly selected images from the *FU-PN2V Convallaria*, *FU-PN2V Mouse actin*, *FU-PN2V Mouse nuclei*, and *W2S Ch.1 (avg1)* datasets. We compare SELF2SELF to DIVNOISING when $(i)$ trained on the same randomly chosen image from the respective dataset, and $(ii)$ when DIVNOISING was trained on the entire dataset and applied on the respective randomly selected image. Within a generous time limit of 10 hours for training per image, DIVNOISING still outperforms SELF2SELF in measured PSNR performance while requiring about 7 times less GPU memory (see Appendix A.14 and Appendix Table 2). Note that the application of SELF2SELF to an entire dataset containing 100 images would require 1000 hours of cumulative training time, while an overall 10 hour training of DIVNOISING on the entire dataset is sufficient to denoise all contained images. The performance on the natural image benchmark dataset BSD68 (Roth & Black, 2005) is shown in Fig. 26 and discussed in Appendix A.10. Additional qualitative results for all datasets can be found in Appendix A.9. A discussion on the accuracy of the posterior modeled by DIVNOISING can be found in Appendix A.11.

**Downstream Processing: OCR.** In Fig. 4 we show how Optical Character Recognition (OCR) applications might benefit from diverse denoising. While regular denoising approaches predict poor compromises that would never be seen in clean text, DIVNOISING can generate a diverse set of rather plausible denoised solutions. While our MAP estimates clean up most such problems, occasional mistakes cannot be avoided, *e.g.* changing "hunger" to "hungor" (see Fig. 4). Diverse denoising solutions obtained by clustering typically correspond to plausible alternative interpretations. It stands to reason that OCR systems can benefit from having access to diverse interpretations.

**Downstream Processing: Instance Cell Segmentation.** We demonstrate how diverse denoised images generated with DIVNOISING can help to segment all cells in the *DenoiSeg Flywing* data.

While methods to generate diverse segmentations do exist (Kohl et al., 2018; 2019), they require ground truth segmentation labels during training. In contrast, we use a simple and fast downstream segmentation pipeline $c(\mathbf{x})$ based on local thresholding and skeletonization (see Appendix A.6 for details) and apply it to individual samples $(\mathbf{s}^1 \ldots \mathbf{s}^K)$ predicted by DIVNOISING to derive segmentations $(\mathbf{c}^1 \ldots \mathbf{c}^K)$. We explore two label fusion methods to combine the individual results and obtain an improved segmentation. We do: $(i)$ use *Consensus (BIC)* (Emre Akbas et al., 2018) and $(ii)$ create a pixel-wise average of $(\mathbf{c}^1 \ldots \mathbf{c}^K)$, followed by again applying our threshold based segmentation procedure on this average, calling it *Consensus (Avg)*.

For comparison, we also segment $(i)$ the low SNR input images, $(ii)$ the original high SNR images, and $(iii)$ the MMSE solutions of DIVNOISING. Figure 5 and Appendix Fig. 11 show all results of our instance segmentation experiments. It is important to note that segmentation from even a single DIVNOISING prediction outperforms segmentations on the low SNR image data quite substantially. We observe that label fusion methods can, by utilizing multiple samples, outperform the MMSE estimate, with *Consensus (Avg)* giving the best overall results (see Appendix Fig. 11).

# 7    DISCUSSION AND CONCLUSION

We have introduced DIVNOISING, a novel unsupervised denoising paradigm that allows us, for the first time, to generate diverse and plausible denoising solutions, sampled from a learned posterior. We have demonstrated that the quality of denoised images is highly competitive, typically outperforming the unsupervised state-of-the-art, and at times even improving on supervised results.[1]

DIVNOISING uses a lightweight fully convolutional architecture. The success of Deep Image Prior (Ulyanov et al., 2018) shows that convolutional neural networks are inherently suitable for image denoising. Yokota et al. (2019) reinforce this idea and Tachella et al. (2020) additionally hypothesize that a possible reason for the success of convolutional networks is their similarity to non-local patch based filtering techniques. However, the overall performance of DIVNOISING is not merely a consequence of its convolutional architecture. We believe that the novel and explicit modeling of imaging noise in the decoder plays an essential role. This becomes evident when comparing our results to other convolutional baselines (including Deep Image Prior and fully convolutional VAEs), which do not perform as well as DIVNOISING on any of the datasets we used. Additionally, we observe that incorrect noise models consistently lead to inferior results (see Appendix A.7).

We find that DIVNOISING is suited particularly well for microscopy data or other applications on a limited image domain. In its current form it works less well on collections of natural images (see Appendix A.10). This might not be very surprising, as we are training a generative model for our image data and would not expect to be capturing the tremendously diverse domain of natural photographic images with the comparatively tiny networks used in our experiments (see Appendix A.3). For microscopy data, instead, the diversity between datasets can be huge. Images of the same type of sample, acquired using the same experimental setup, however, contain many resembling structures of lesser overall diversity (they are from a limited image domain). Nevertheless, the stunning results we achieve suggest that DIVNOISING will also find application in other areas where low SNR limited domain image data has to be analyzed. Next to microscopy, we can think of astronomy, medical imaging, or limited domain natural images such as faces or street scenes. Additionally, follow up research will explore larger and improved network architectures, able to capture more complex DIVNOISING posteriors on datasets covering larger image domains.

While we constrained ourselves to the standard per-pixel noise models in this paper, the DIVNOISING approach could in principle also work with more sophisticated higher level image degradation models, as long as they can be probabilistically described. This might include diffraction, blur, or even compression and demosaicing artefacts.

Maybe most importantly, DIVNOISING can not only produce competitive and diverse results, but these results can also be leveraged for downstream processing. We have seen that cell segmentation can be improved and that clustering our results provides us with meaningful alternative interpretations of the same data (see Fig. 4). We believe that this is a highly promising direction for many applications, as it provides us with a way to account for the uncertainty introduced by the imaging process. We are looking forward to see how DIVNOISING will be applied and extended by the community, showcasing the true potential and limitations of this approach.

---

[1] Supervised methods using perfect GT will outperform DIVNOISING, but GT data is at times not perfect.

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

# A  APPENDIX

## A.1  INTRINSICALLY NOISY MICROSCOPY DATA

We use public microscopy datasets which show realistic levels of noise, introduced by the respective optical imaging setups. The *FU-PN2V Convallaria* (Krull et al., 2020; Prakash et al., 2020) data, consists of 100 noisy calibration images (intended to generate a noise model), and 100 images of size $1024 \times 1024$ showing a noisy Convallaria section. The *FU-PN2V Mouse nuclei* (Prakash et al., 2020) data is composed of 500 noisy calibration images and 200 noisy images of size $512 \times 512$ showing labeled cell nuclei. The *FU-PN2V Mouse actin* (Prakash et al., 2020) data from the same source consists of 100 noisy calibration images and 100 noisy images of size $1024 \times 1024$ of the same sample, but labeled for the protein actin. Finally, we use all 3 channels of 2 noise levels (avg1 and avg16) of the *W2S* (Zhou et al., 2020) data. For each channel, corresponding high quality (ground truth) images are available. Each channel's training and test sets consist of 80 and 40 images, respectively. All images are $512 \times 512$ pixels in size.

## A.2  DATA EXPOSED TO SYNTHETIC NOISE

We use the well known *MNIST* (LeCun et al., 1998) as well as the *KMNIST* (Clanuwat et al., 2018) dataset showing $28 \times 28$ images of handwritten digits and phonetic letters of hiragana, respectively. Both datasets contain 60000 training examples and 10000 test examples. Onto both datasets we added pixel-wise independent Gaussian noise with $\mu = 0$ and $\sigma = 140$. As a third text-based dataset we rendered the freely available *eBook* "The Beetle" (Marsh, 2004) and extracted 40800 image patches of size $128 \times 128$. We separated 34680 patches for training and 6120 patches for validation, and added pixel-wise independent Gaussian noise with $\mu = 0$ and $\sigma = 255$. Additionally, we use three datasets from microscopy. The *DenoiSeg Mouse* (Buchholz et al., 2020) data, showing cell nuclei in the developing mouse skull, consists of 908 training and 160 validation images of size $128 \times 128$, with additional 67 images of size $256 \times 256$ for testing. Two noisy datasets were created with this data, one by exposing all images to pixel-wise independent Gaussian noise with $\mu = 0$ and $\sigma = 20$ and another one by first applying poisson noise with $\lambda = 1$ followed by adding gaussian noise with $\mu = 0$ and $\sigma = 10$ followed by randomly changing $3\%$ of pixels to either 0 or 255. This dataset is called Mouse s&p in Table 1. The *DenoiSeg Flywing* (Buchholz et al., 2020) data is showing membrane labeled cells in a fly wing, consisting of 1428 training and 252 validation patches of size $128 \times 128$, with additional 42 images of size $512 \times 512$ for testing. We exposed this data to pixel-wise independent Gaussian noise with $\mu = 0$ and $\sigma = 70$ to create a synthetic low SNR version. All original datasets are 8-bit. Lastly, we randomly select 500 images of size $384 \times 286$ from BioID Face recognition database (noa) and corrupt them with pixel-wise independent Gaussian noise with $\mu = 0$ and $\sigma = 15$. We use $340, 60$ and 100 images for training, validation and test respectively.

## A.3  TRAINING AND NETWORK DETAILS

Here, we provide additional details about the network architecture and training parameters used throughout the main manuscript. For all DIVNOISING experiments, we use rather lightweight depth 2 and depth 3 VAE architectures (see Appendix Figs. 6 and 7, respectively). All networks use a single input channel and 32 feature channels in the first network layer except for the network trained on mouse s&p dataset which uses 96 feature channels in the first network layer. We use two $3 \times 3$ convolutions (with padding 1), each followed by ReLU activation, followed by a $2 \times 2$ max pooling layer. After each such downsampling step, we double the number of feature channels. For all experiments we use a network architecture of depth 2 (with 2 down/upsampling steps). The only exceptions are our experiments on *DenoiSeg Flywing* and *eBook* data, for which we use a depth 3 architecture (with 3 down/upsampling steps). In total, our depth 2 networks have only around $200k$ parameters and depth 3 networks have around $700k$ parameters.

While we generally use a VAE bottleneck of 64 latent space feature dimensions for each pixel of the image (after encoding), for the small $28 \times 28$ MNIST and KMNIST images we use only 8 such latent space dimensions.

We consistently use 8-fold data augmentation (rotation and flipping) in all experiments. All networks are trained with a batch size of 32 and an initial learning rate of 0.001. The learning rate is multiplied by 0.5 if the validation loss does not decrease for 30 epochs.

For all datasets other than MNIST and KMNIST, we extract training patches of size $128 \times 128$, and separate 15% of all patches for validation. We set the maximum number of epochs such that approximately 22 million steps are performed, and in each epoch the entire training data is being fed. Training is terminated if the validation loss does not decrease by at least $10^{-6}$ over 100 epochs.

For *DenoiSeg Flywing* we observed KL *vanishing* and solved it via *Annealing* within the first 15 epochs (Bowman et al., 2015).

The fully unsupervised DIVNOISING decoder directly predicts the signal and the noise variance per pixel where the variance is constrained to linearly depend on the signal (See Section 5). To avoid numerical problems and ensure that the predicted variance always remains positive, we allow the user to set a minimum allowed variance/standard deviation $\sigma^2_{\min}/\sigma_{\min}$, and enforce this by clamping the predicted values. Note that a viable choice for this parameter depends on the intensity range of the dataset. We use the following values: For all *FU-PN2V* datasets $\sigma_{\min} = 50$, for *DenoiSeg Flywing* and *DenoiSeg Mouse* datasets $\sigma_{\min} = 3$, for *DenoiSeg Mouse s&p* dataset $\sigma_{\min} = 1$, for *BioID Face* dataset $\sigma_{\min} = 15$, for *W2S avg 1* datasets $\sigma_{\min} = 25$ and for *W2S avg 16* datasets $\sigma_{\min} = 3$.

**Run Time and Hardware Requirements.**

DIVNOISING using light weight fully convolutional networks (see Appendix Figs. 6 and 7) runs on relatively cheap computational budget. Our depth 2 networks trained for all experiments requires about 1.8 GB GPU memory and our depth 3 networks roughly 5 GB GPU memory on a *NVIDIA TITAN Xp* GPU. The training time varied from $5 - 12$ hours on average depending on the dataset.

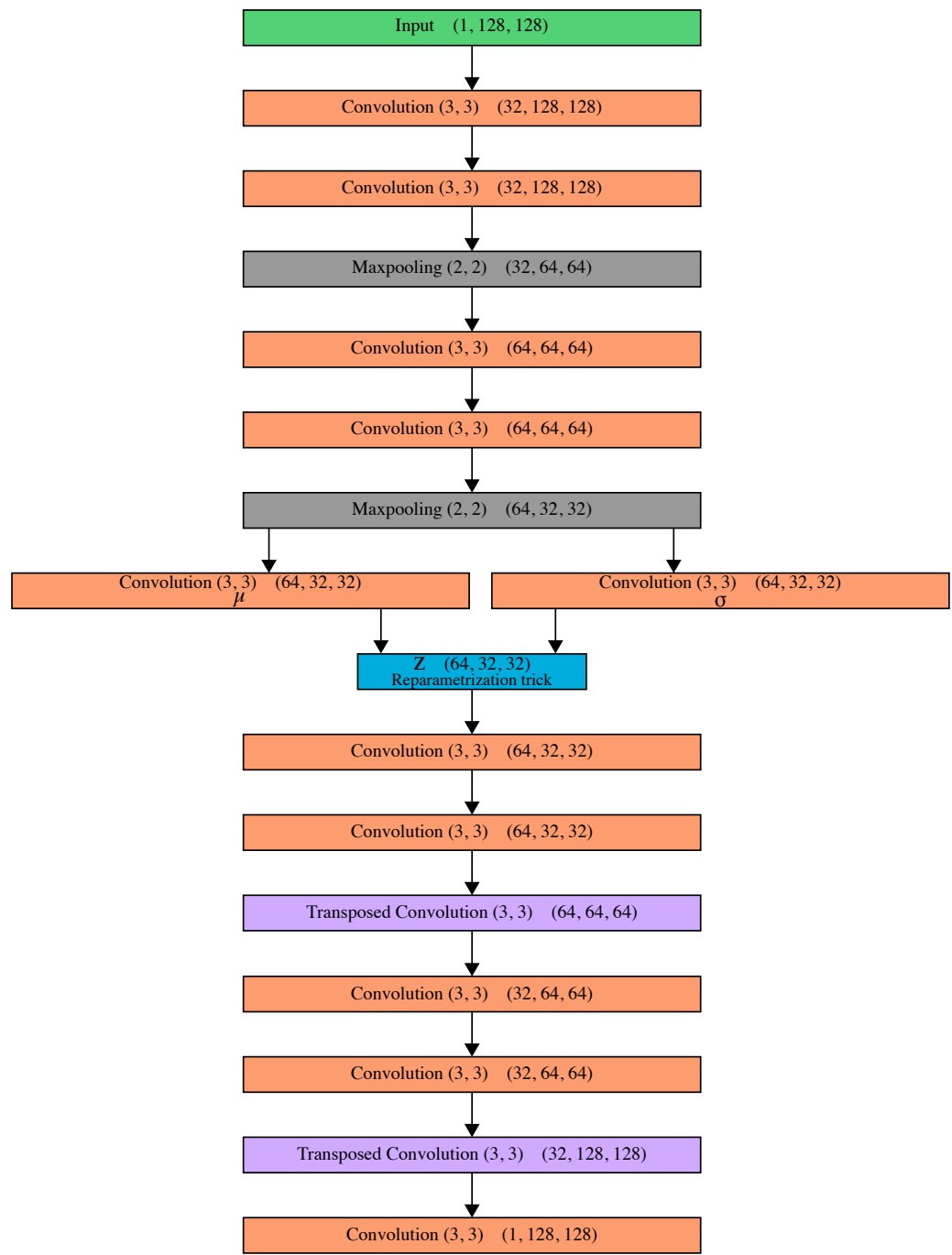

Figure 6: **The fully convolutional architecture used for depth 2 networks.** We show the depth 2 DIVNOISING network architecture used for *FU-PN2V Convallaria, FU-PN2V Mouse nuclei, FU-PN2V Mouse actin*, all *W2S* channels and *DenoiSeg Mouse* datasets. These networks count about $200k$ parameters and have a GPU memory footprint of approximately 1.8GB on a *NVIDIA TITAN Xp*.

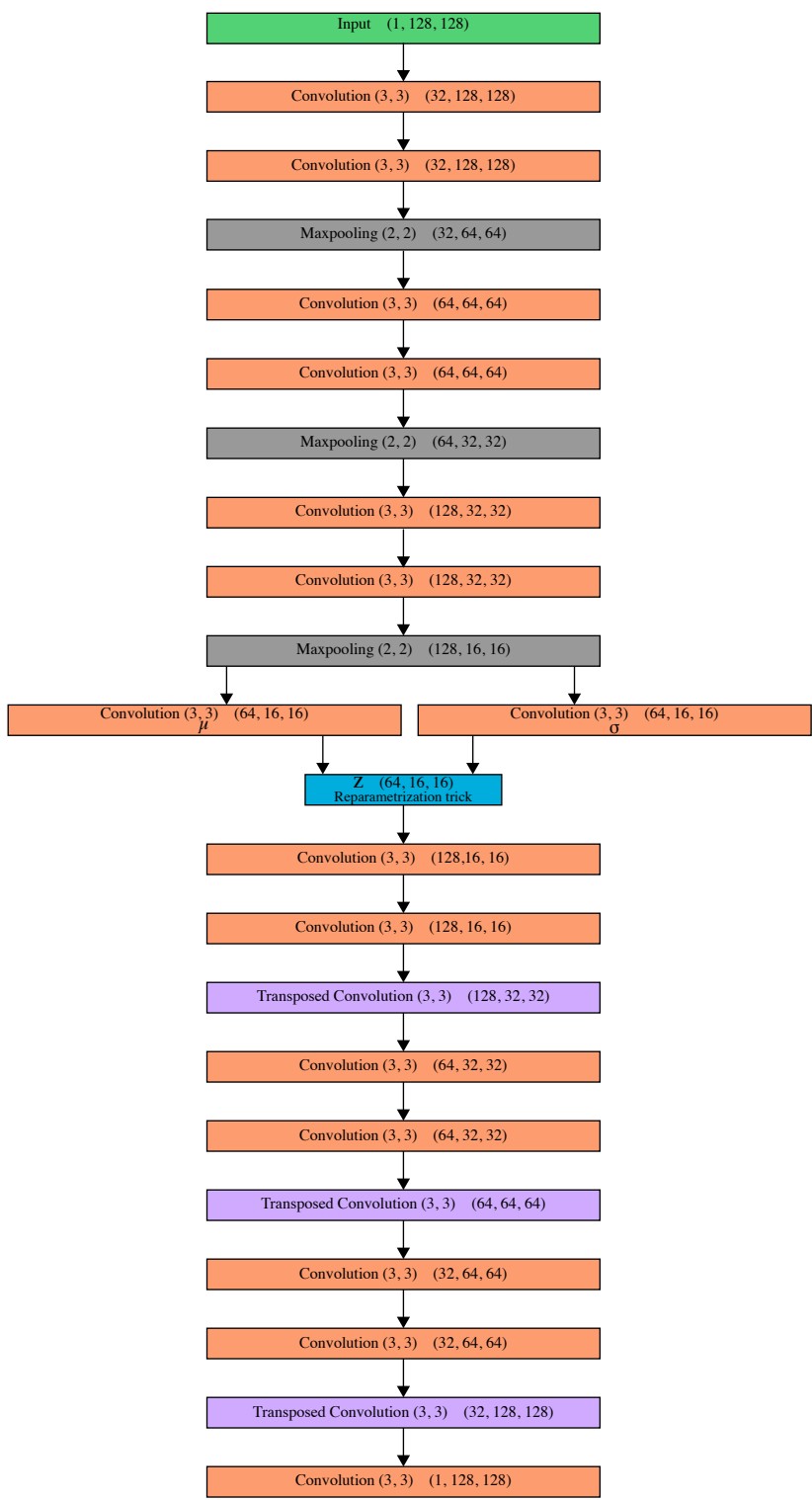

Figure 7: **The fully convolutional architecture used for depth** 3 **networks.** We show the depth 3 DIVNOISING network architecture used for *DenoiSeg Flywing* and *eBook* datasets. These networks count about $700k$ parameters and have a GPU memory footprint of approximately 5GB on a *NVIDIA TITAN Xp*.

### A.4 Clustering of Solutions and Deriving the MAP Estimate

Here we provide additional details on how the cluster centers and the approximate MAP estimate of Fig. 4 (see main text) were found. We first drew 10000 sampled images from the approximate posterior as described in Section 4 of the main text. We then performed mean shift (Cheng, 1995) clustering (using the existing *scipy* implementation) on the cropped image region shown in the figure. We set a *bandwidth* of 800 and the the *maximum number of iterations* to 20, and used the 100 first samples of DivNoising as seeds. We finally show 9 of the resulting cluster centers in the figure.

To produce the MAP estimate, we employ a similar strategy. In order to find the mode of the sampled distribution efficiently, we assume that dependencies in the predicted samples should be local. This assumption is valid, since our network only has only a finite receptive field for each predicted pixel. Hence, we apply mean shift algorithm on locally overlapping regions. We use a window size of $10 \times 10$ pixels with an overlap of 3 pixels in $x$ and $y$. On each such region, the mean shift algorithm is executed repeatedly with decreasing bandwidth, always using the latest result as new seed. We start by using the sample mean as seed and with an initial *bandwidth* of 200. After each iteration the bandwidth is decreased by a factor of 0.9, until it drops below 100.

Similar results should also be achievable by applying mean shift algorithm on the entire image. But since samples will differ at any location in the image, this global approach would require an excessively large number of DivNoising samples.

### A.5 Generating Images with DivNoising Models

Just as with a vanilla VAE (see Section 4 in the main text), we can use a trained DivNoising VAE to synthesise images of structures resembling the training data. To achieve this, we sample from the normal distribution $\mathbf{z}^k \sim p(\mathbf{z})$ and process each sample with the decoder network $\mathbf{s}^k = g_\theta(\mathbf{z}^k)$. We show such generated images in comparison to real crops from the test data in Appendix Figs. 8 to 10. We see that the images appear most plausible for local structures, indicating that the small networks we use in this work are not capable of capturing larger structural features in the given data.

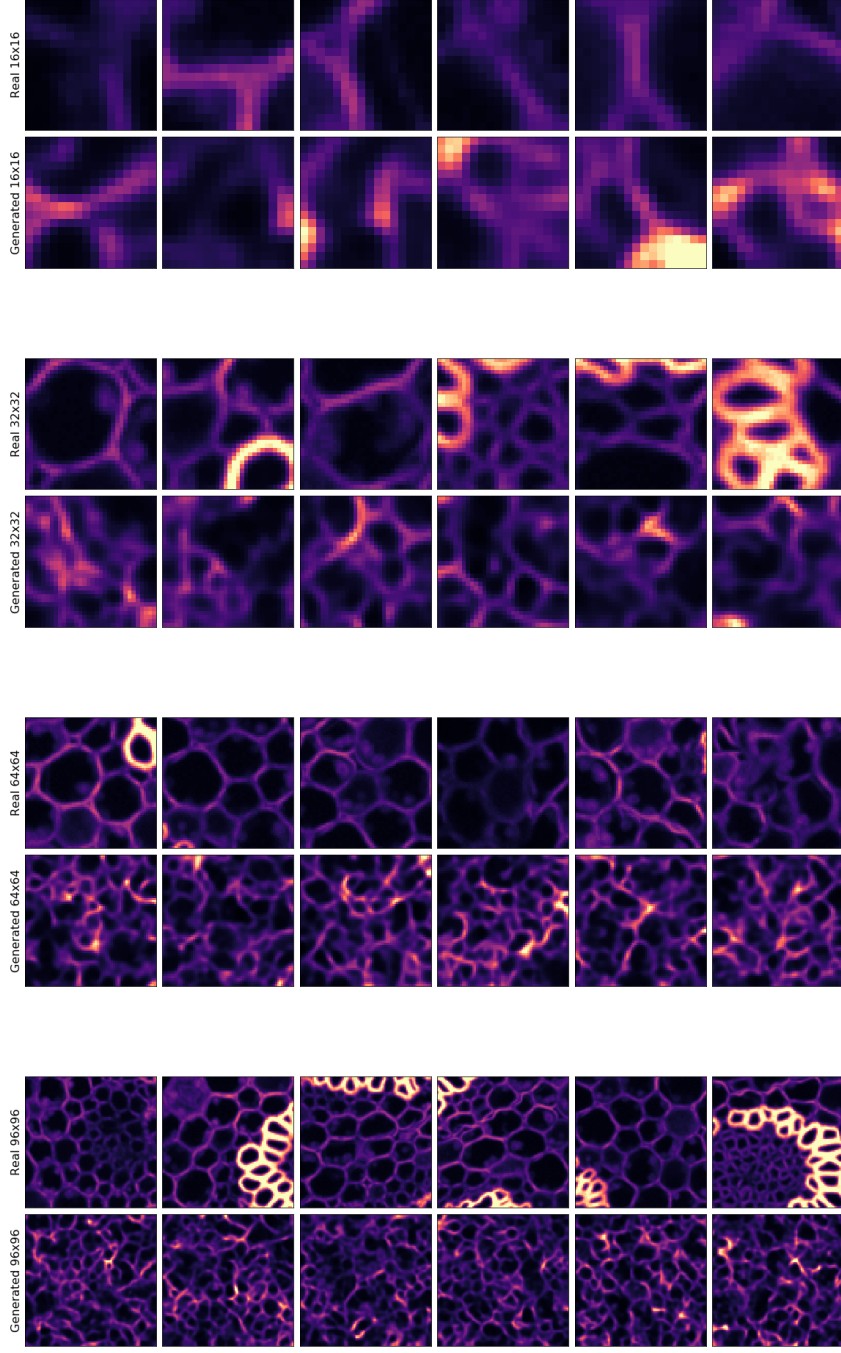

Figure 8: **Generating synthetic images with the DIVNOISING VAE for the *FU-PN2V Convallaria* dataset Krull et al. (2020); Prakash et al. (2020).** DIVNOISING can also be used to generate images by sampling from the unit normal distribution $p(\mathbf{z})$ and then using the decoder to produce an image. Here, we compare generated images and randomly cropped real images. We show images of different resolutions to see how well the VAE captures structures at different scales. The VAE we use for denoising is only able to realistically capture small local structures. Note that the network we use is quite shallow (see Appendix Fig. 6).

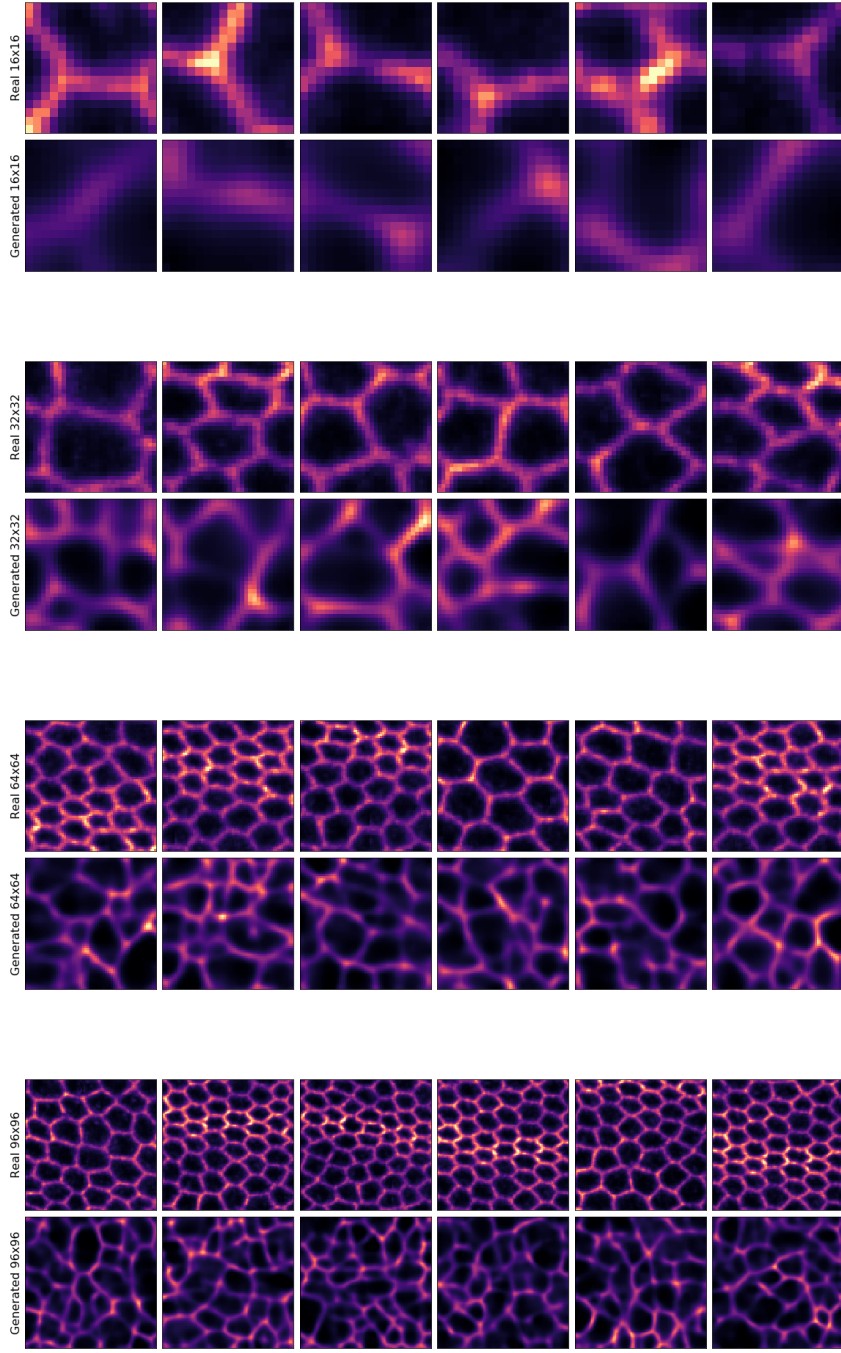

Figure 9: **Generating synthetic images with the DIVNOISING VAE for the *DenoiSeg Fly-wing* Buchholz et al. (2020) dataset.** DIVNOISING can also be used to generate images by sampling from the unit normal distribution $p(\mathbf{z})$ and then using the decoder to produce an image. Here, we compare generated images and randomly cropped real images. We show images of different resolutions to see how well the VAE captures structures at different scales. Note that the network (see Appendix Fig. 7) we use is a bit deeper compared to Supplementary Fig. 8. This VAE captures larger structures a little better but struggles to produce crisp high frequency structures. This is likely a consequence of the increased depth of the used network.

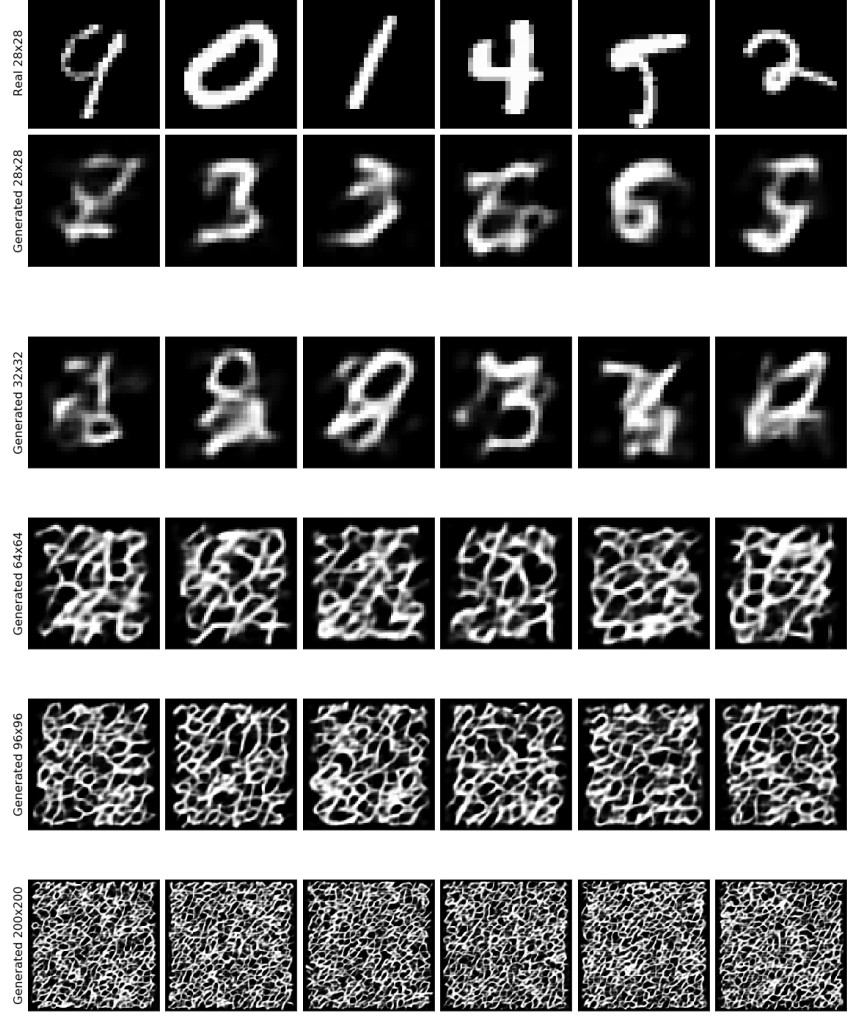

Figure 10: **Generating synthetic images with the DIVNOISING VAE for the MNIST LeCun et al. (1998) dataset.** DIVNOISING can also be used to generate images by sampling from the unit normal distribution $p(\mathbf{z})$ and then using the decoder to produce an image. Here, we compare generated images and random ground truth images. Our fully convolutional architecture allows us to generate images of different sizes (despite all input images being only of size $28 \times 28$).

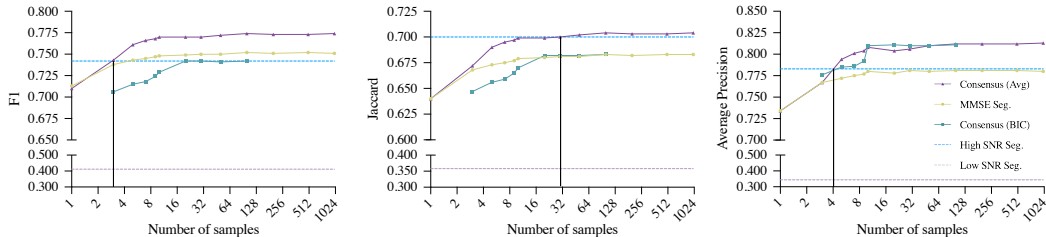

Figure 11: **DIVNOISING enables downstream segmentation.** Evaluation of segmentation results (using the F1 score (Van Rijsbergen, 1979), Jaccard score (Jaccard, 1901) and Average Precision (Lin et al., 2014). On the x-axis we plot the number of DIVNOISING samples used. The performance of BIC is only evaluated up to 100 samples because we limited run-time to 30 minutes). Remarkably, *Consensus (Avg)* using only 30 DIVNOISING segmentation labels, outperforms segmentations obtained from high SNR images.

## A.6    INSTANCE CELL SEGMENTATION

Here, we provide additional details regarding the downstream segmentation task described in Section 6 of the main text. We used the first 21 images in the test set of *DenoiSeg Flywing* for our analysis.

Given an input image, our segmentation pipeline consists of $(i)$ generating segmentation masks using local thresholding with a mean filter of radius 15, followed by $(ii)$ skeletonizing the space between these masks, followed by $(iii)$ connected component analysis to obtain instance segmentation.

Using this pipeline, we generated segmentation for the noisy (low SNR) images, ground truth (high SNR) images, as well as for the DIVNOISING MMSE estimate (obtained by averaging 1000 sampled denoised images).

We also apply the above described pipeline for each of the 1000 DIVNOISING samples separately to serve as input for the two label fusion methods, namely $(i)$ *Consensus (BIC)*, and $(ii)$ *Consensus (Avg)*. For the latter label fusion method we skip the connected component analysis and directly average the thresholded and skeletonized images. To obtain the final result, we again apply the full segmentation pipeline described above to this average image.

All segmentations were obtained with the open source image analysis software Fiji (Schindelin et al., 2012).

The quantitative results illustrating the benefit of diverse segmentation for label fusion methods is shown in Appendix Fig. 11.

## A.7    THE RELATIVE IMPORTANCE OF THE KL LOSS COMPONENT

We can generalize our DIVNOISING training loss as a weighted combination of a modified reconstruction loss (see Section 5 in the main text) and KL divergence loss, where the two loss components are weighted equally. Following the exposition in (Higgins et al., 2017), we explore the effect of weighting the KL loss component during training with a factor $\beta$. Our modified training loss thus becomes

$$\mathcal{L}_{\phi,\theta}(\mathbf{x}) = \mathcal{L}^{\mathrm{R}}_{\phi,\theta}(\mathbf{x}) + \beta \mathcal{L}^{\mathrm{KL}}_{\phi}(\mathbf{x}), \tag{5}$$

where setting $\beta = 1$ gives our DIVNOISING setup described in Section 5 in the main text. Note that increasing or reducing $\beta$, *i.e.* changing the relative importance of the reconstruction loss, is equivalent to using a wider or narrower noise model, such as a Gaussian noise model with larger or smaller standard deviation $\sigma$. We can thus interpret above results as the effect of using a mismatched noise model that is either too wide or too narrow.

**Effect of $\beta$ on Denoising Quality.**  We investigated the effect of $\beta$ on the denoising ability of DIVNOISING network with the *DenoiSeg Flywing* dataset. As illustrated in Appendix Fig. 12a, $\beta = 1$ gives the optimal results for the MMSE estimate (obtained by averaging 1000 samples). Both regimes, $\beta > 1$ and $\beta < 1$, yield sub-par denoising performance.

**Effect of $\beta$ on Diversity of Denoised Samples.** We introduce a simple new metric, called *standard deviation PSNR*, to quantify the diversity of denoised results obtained as a function of $\beta$. For a given noisy image $x$ and given a set of denoised samples $S_x$, we compute the PSNR of each sample $a \in S_x$ with respect to the corresponding ground truth image $s$. This yields a vector of PSNR values $\mathbf{v}$ where

$v_i = PSNR(s, a_i)$, for $v_i \in \mathbf{v}$. Standard deviation PSNR for the noisy image $x$ is then defined as the standard deviation of elements in the vector $v$. Appendix Fig. 12b reports the average of standard deviation PSNR obtained for $42$ test images of the *DenoiSeg Flywing* dataset. The higher the beta, the higher is the standard deviation PSNR indicating higher diversity. Qualitative results presented in Appendix Fig. 13 show that with $\beta > 1$, there is an increased diversity at the bigger image scales (e.g. diverse predictions of cell membranes), and generated denoised images appear smoother than those observed in real data. Setting $\beta < 1$ reduces diversity and introduces grainy artefacts, thereby yielding poor reconstructions. Note that $\beta = 1$ gives the best results in terms of PSNR of MMSE while maintaining a fair level of diversity.

## A.8  HOW DOES NOISE AFFECT THE DIVERSITY OF DIVNOISING SAMPLES?

We quantified how the diversity of DIVNOISING samples changes with the amount of noise present in the original dataset. Increased level of noise introduces additional uncertainty about the true signal, hence we would expect this to lead to increasingly diverse samples.

To test this hypothesis, we choose the *DenoiSeg Flywing* dataset and inject pixel wise independent gaussian noise of mean 0 and standard deviations $\sigma = 30, 50$ and 70. We report the standard deviation PSNR diversity metric, introduced in Appendix Section A.7, for all three noise levels. As demonstrated in Appendix Fig. 12c, the higher the noise level, the more diverse the DIVNOISING samples become, thereby confirming our hypothesis.

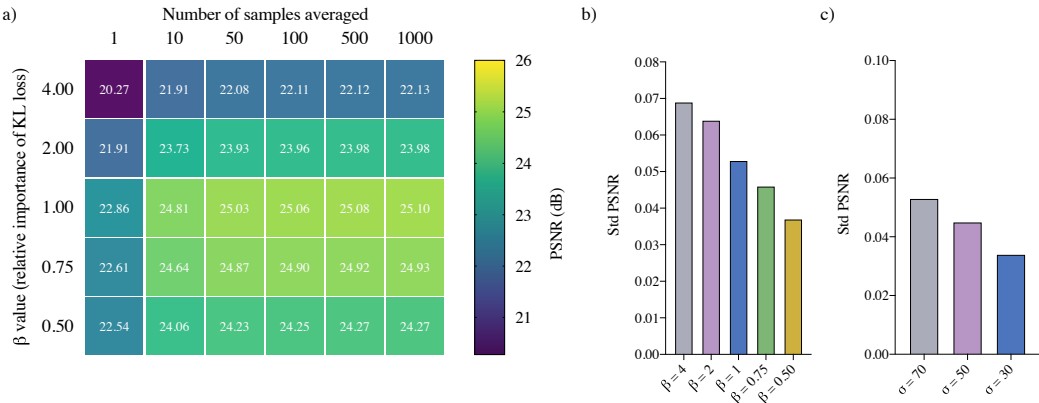

Figure 12: **Analyzing the denoising quality and diversity of DIVNOISING samples with different factors for the *DenoiSeg Flywing* dataset.** **(a)** The heatmap shows how the quality (PSNR in db) of DIVNOISING MMSE estimate changes with averaging increasingly larger number of samples (numbers shown for 1 run). Unsurprisingly, the more samples are averaged, the better the results get. We also investigate the effect of weighting the KL loss term with a factor $\beta$ (Supplementary Eq. 5) on the quality of reconstruction. We observe that the usual VAE setup with $\beta = 1$ gives the best results in terms of reconstruction quality. Increasing $\beta > 1$ leads to higher diversity at the expense of poor reconstruction (see Appendix Fig. 13.) **(b)** We quantify the denoising diversity achieved with different $\beta$ values in terms of *standard deviation PSNR* (see Appendix section A.7 for details on the metric). We report the average standard deviation of PSNRs over all test images for different values of $\beta$ and observe that the higher $\beta$ values increase the diversity. **(c)** We also investigate the effect of noise on the diversity of denoised DIVNOISING samples by adding pixel wise independent zero mean Gaussian noise of standard deviations 30, 50 and 70. The higher the noise, the more ambiguous the noisy input images are, thus leading to higher diversity.

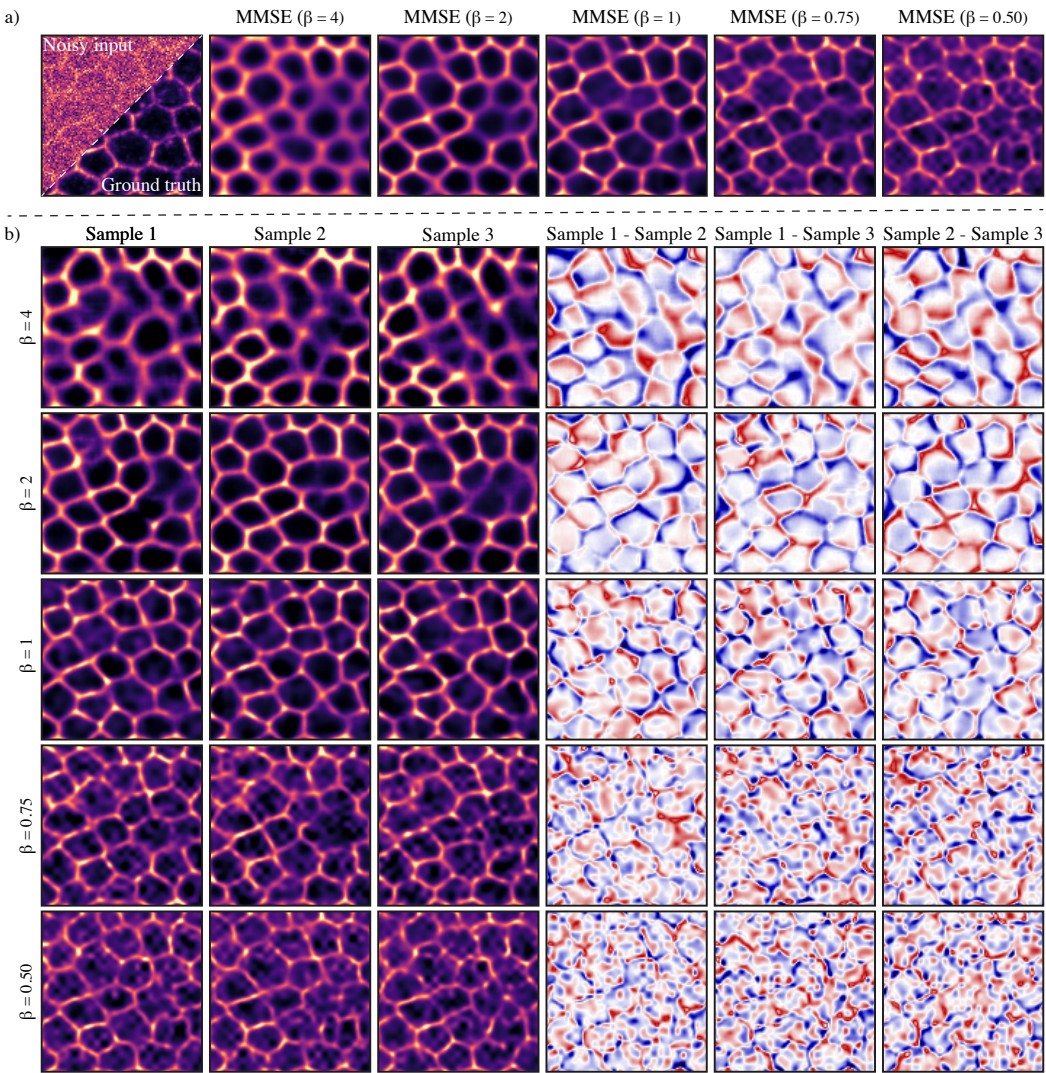

Figure 13: **Qualitative analysis of the effect of weighting KL loss term with factor $\beta$ for *DenoiSeg Flywing* dataset.** **(a)** We show the DIVNOISING MMSE estimate obtained by averaging 1000 samples for all considered $\beta$ values (Supplementary Eq. 5). We observe that the reconstruction quality suffers on either increasing $\beta > 1$ or decreasing $\beta < 1$. Best results (with respect to PSNR) are obtained with $\beta = 1$, as demonstrated in Fig. 12a. **(b)** For each $\beta$ value, we show three randomly chosen DIVNOISING samples as well as difference images. Increasing $\beta > 1$, allows the DIVNOISING network to generate structurally very diverse denoised solutions, while typically leading to textural smoothing. Decreasing $\beta < 1$ generates DIVNOISING samples with overall much reduced structural diversity, introducing reconstruction artefacts/structures at smaller scales.

## A.9 ADDITIONAL RESULTS

**More Qualitative Results.** In addition to the qualitative results presented in Fig. 2 in the main text, here we present more results for each considered dataset in Appendix Figs. A.9-24.

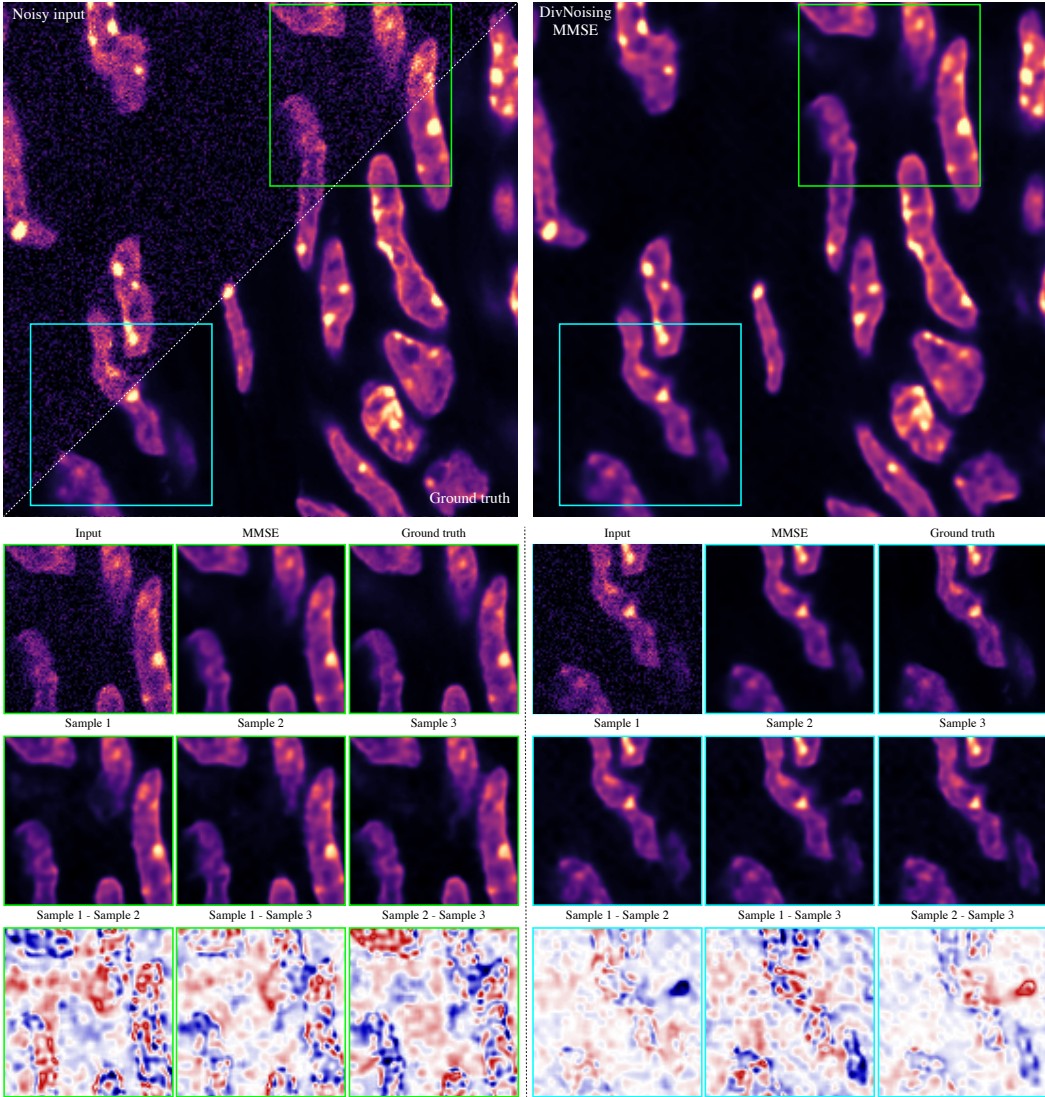

Figure 14: **Additional qualitative results for the *DenoiSeg Mouse* Buchholz et al. (2020) dataset.** Here, we show qualitative results for two cropped regions (green and cyan). The MMSE estimate was produced by averaging 1000 sampled images. We choose 3 samples to display to illustrate the diversity of DIVNOISING results.

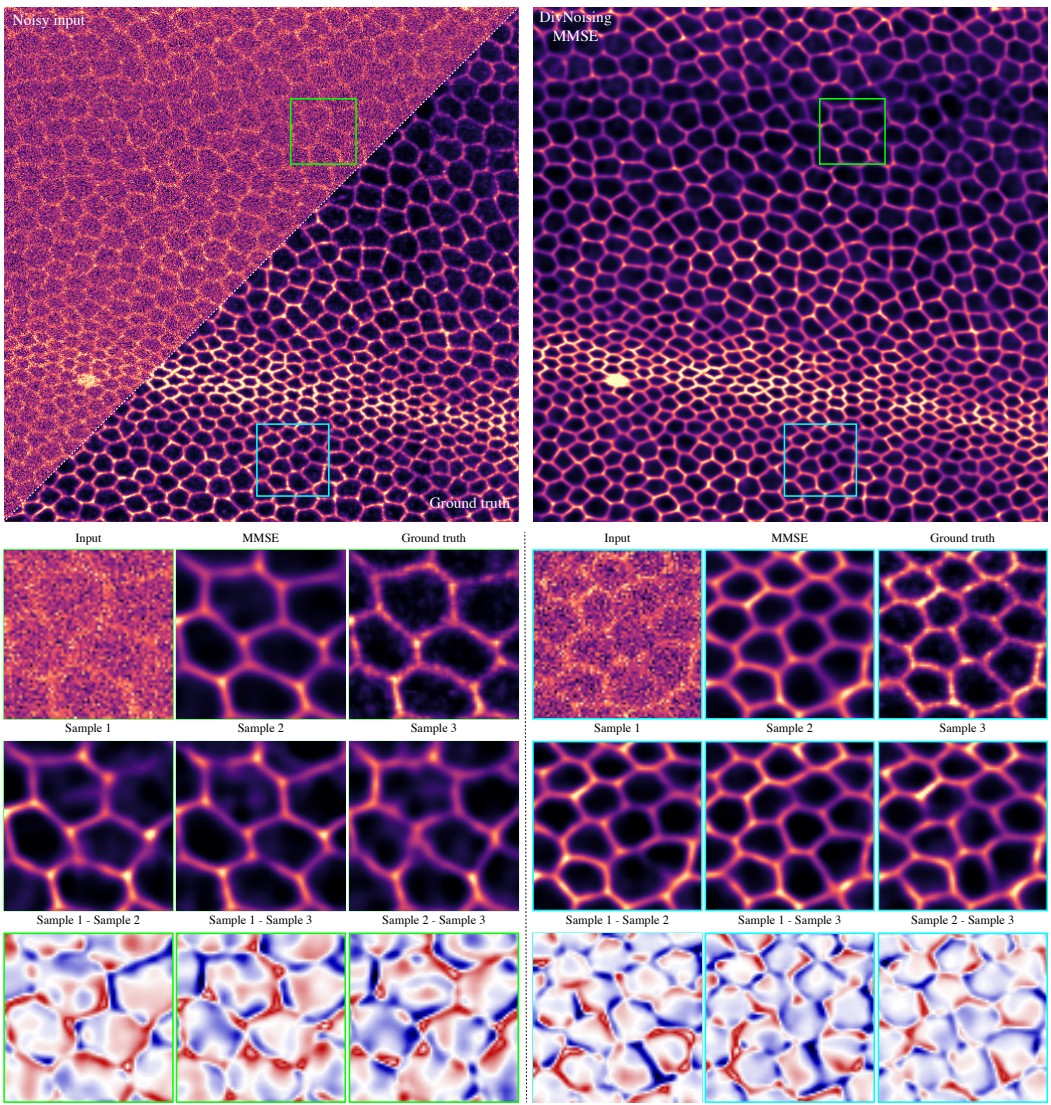

Figure 15: **Additional qualitative results for the *DenoiSeg Flywing* Buchholz et al. (2020) dataset.** Here, we show qualitative results for two cropped regions (green and cyan). The MMSE estimate was produced by averaging 1000 sampled images. We choose 3 samples to display to illustrate the diversity of DIVNOISING results.

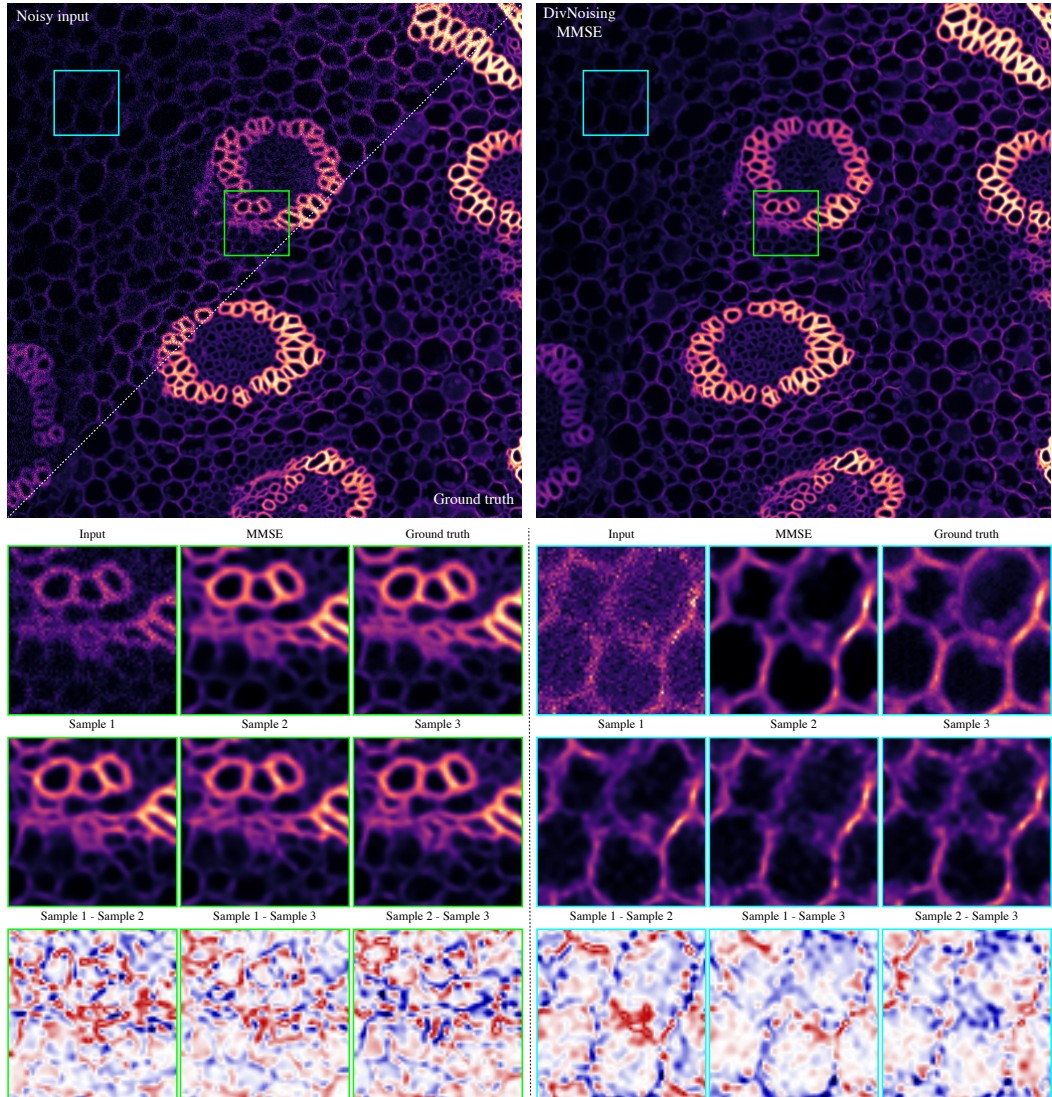

Figure 16: **Additional qualitative results for the *FU-PN2V Convallaria* Krull et al. (2020); Prakash et al. (2020) dataset.** Here, we show qualitative results for two cropped regions (green and cyan). The MMSE estimate was produced by averaging 1000 sampled images. We choose 3 samples to display to illustrate the diversity of DIVNOISING results.

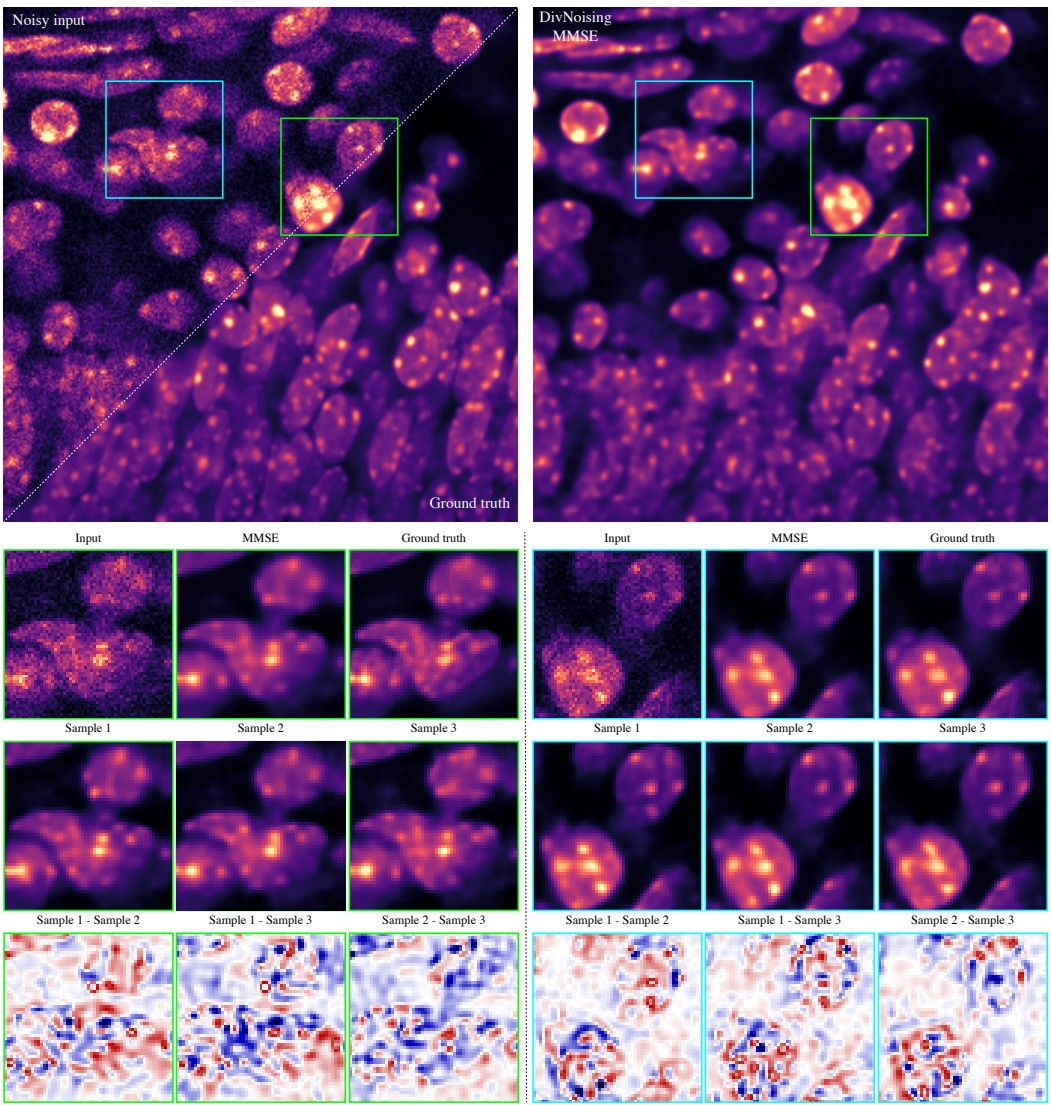

Figure 17: **Additional qualitative results for the *FU-PN2V Mouse nuclei* Prakash et al. (2020) dataset.** Here, we show qualitative results for two cropped regions (green and cyan). The MMSE estimate was produced by averaging 1000 sampled images. We choose 3 samples to display to illustrate the diversity of DIVNOISING results.

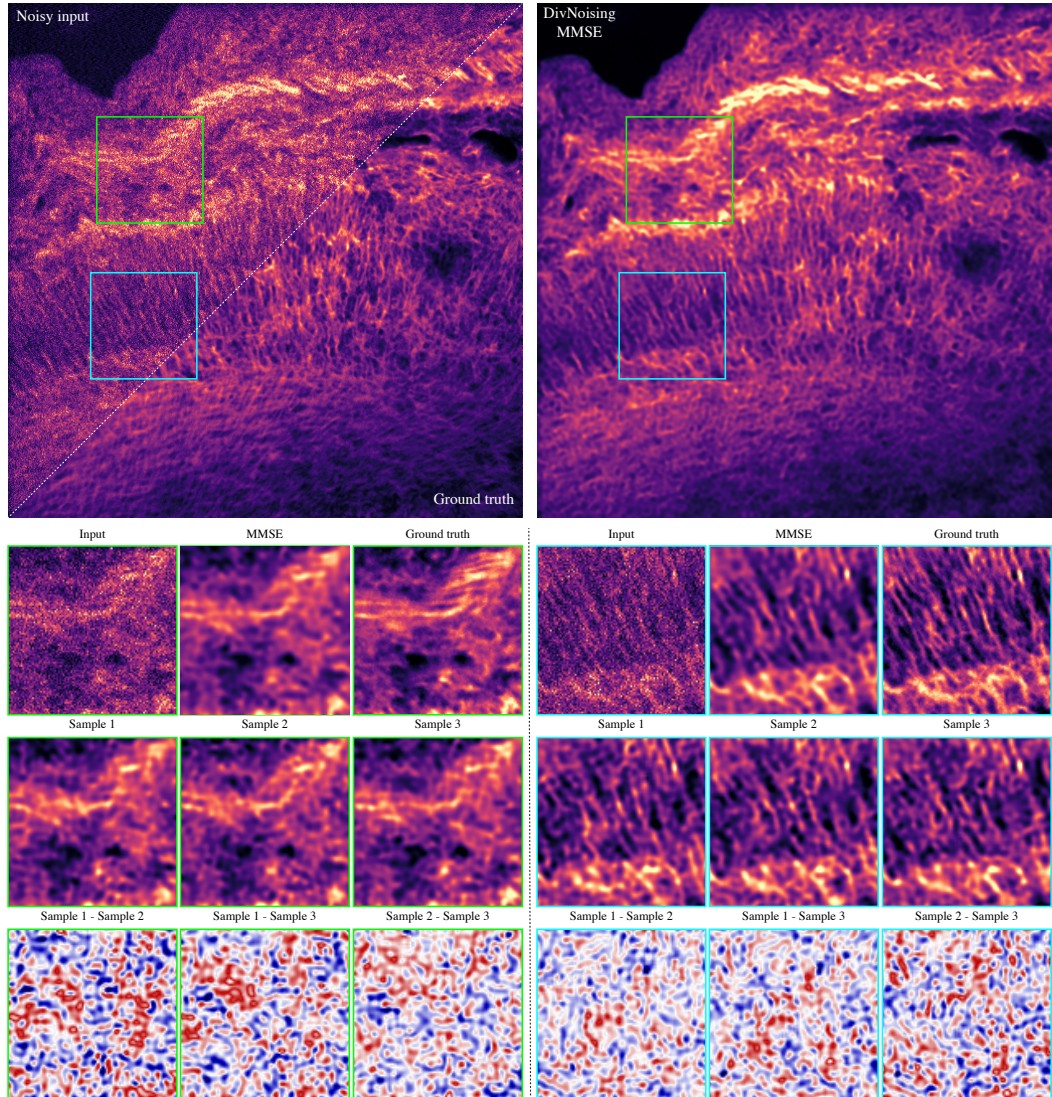

Figure 18: **Additional qualitative results for the *FU-PN2V Mouse actin* Prakash et al. (2020) dataset.** Here, we show qualitative results for two cropped regions (green and cyan). The MMSE estimate was produced by averaging 1000 sampled images. We choose 3 samples to display to illustrate the diversity of DIVNOISING results.

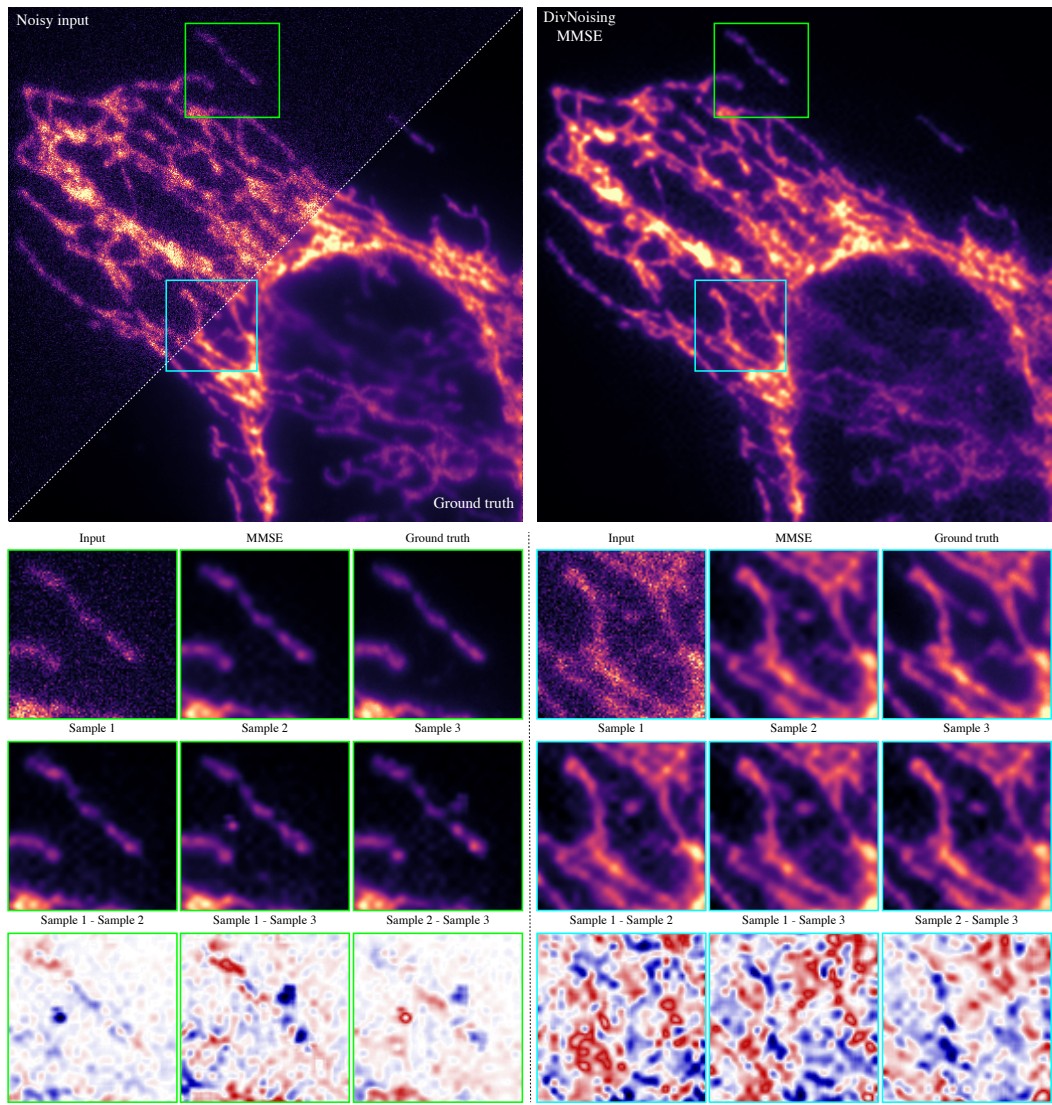

Figure 19: **Additional qualitative results for the *W2S* Zhou et al. (2020) dataset (ch. 0, avg1).** Here, we show qualitative results for two cropped regions (green and cyan). The MMSE estimate was produced by averaging 1000 sampled images. We choose 3 samples to display to illustrate the diversity of DIVNOISING results.

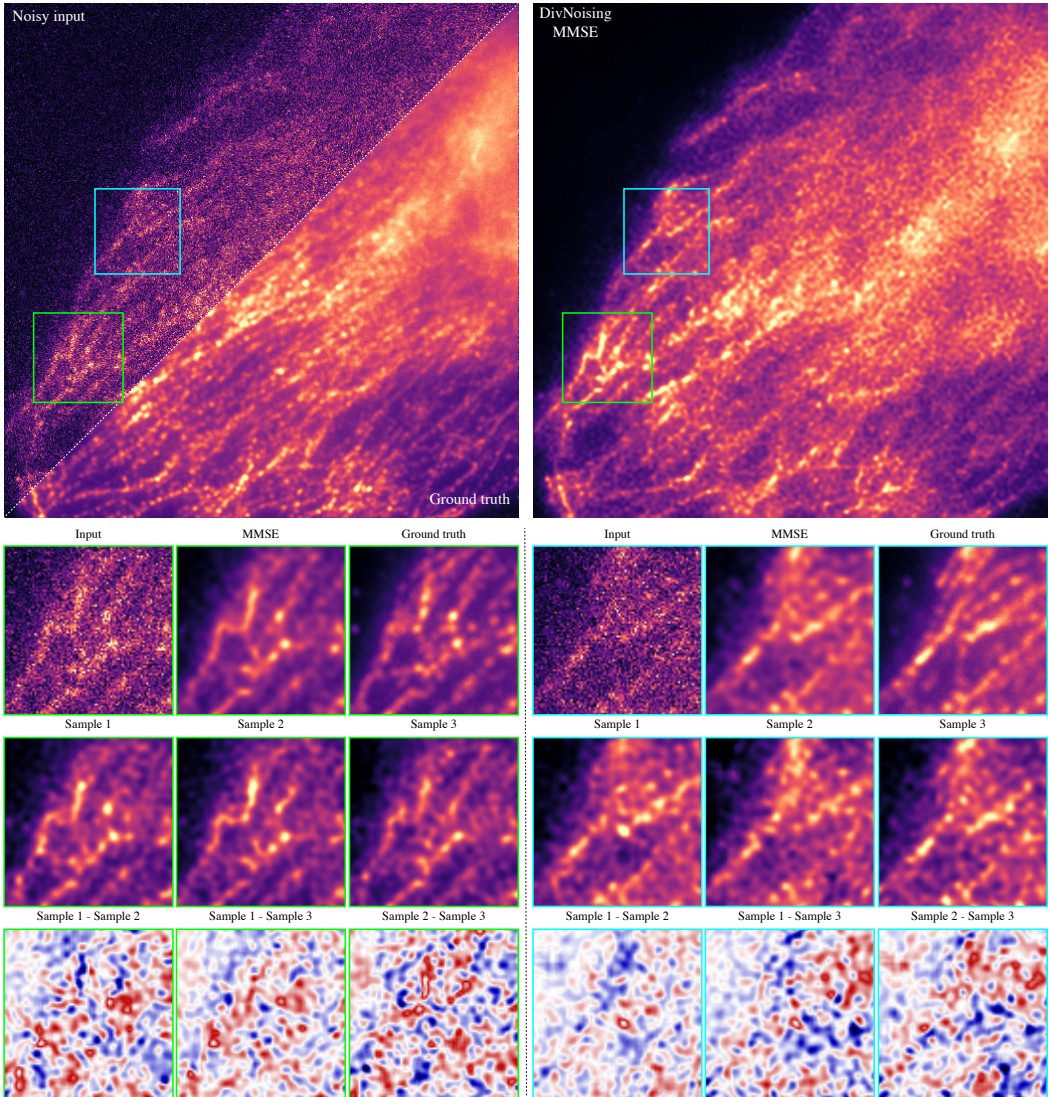

Figure 20: **Additional qualitative results for the *W2S* Zhou et al. (2020) dataset (ch. 1, avg1).** Here, we show qualitative results for two cropped regions (green and cyan). The MMSE estimate was produced by averaging 1000 sampled images. We choose 3 samples to display to illustrate the diversity of DIVNOISING results.

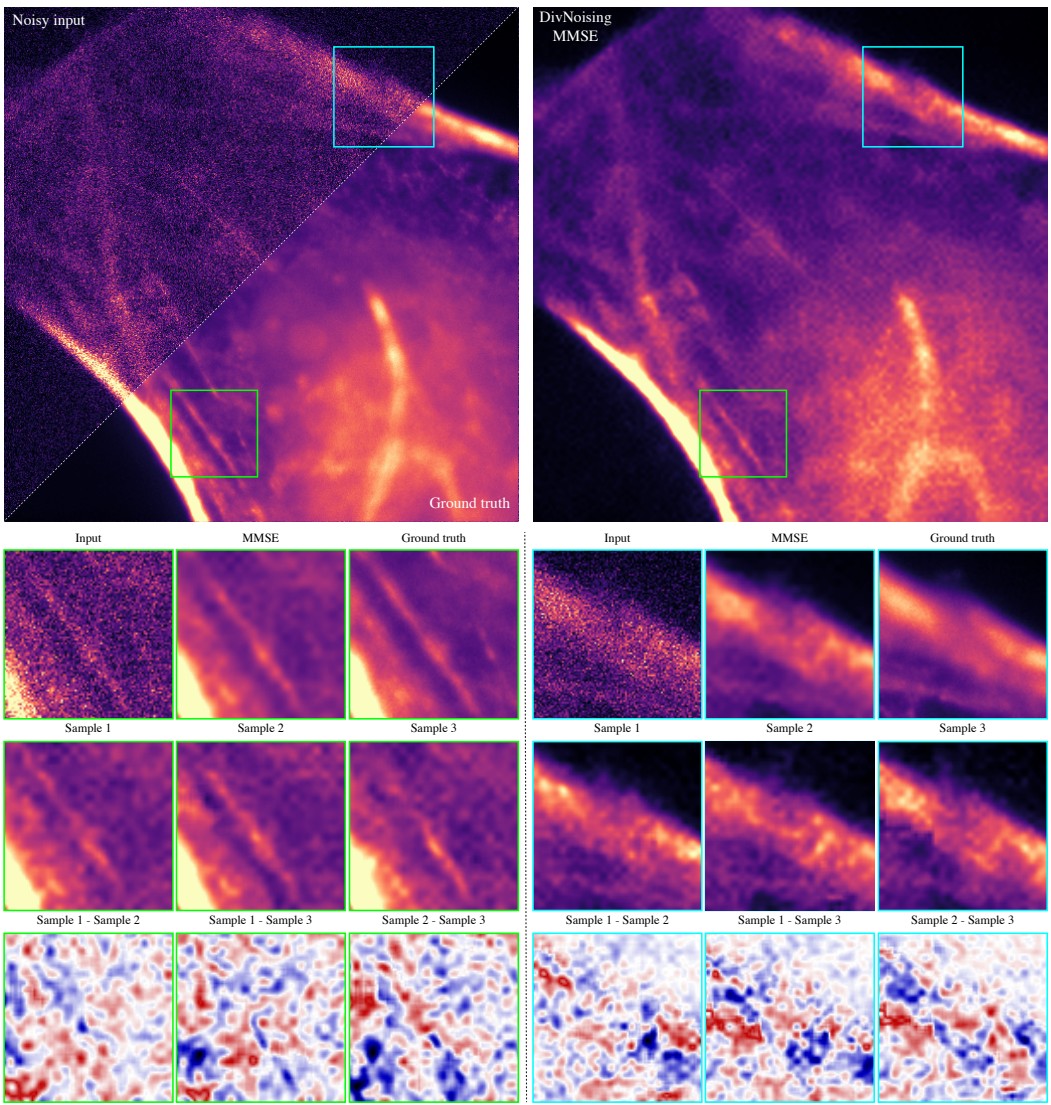

Figure 21: **Additional qualitative results for the *W2S* Zhou et al. (2020) dataset (ch. 2, avg1).** Here, we show qualitative results for two cropped regions (green and cyan). The MMSE estimate was produced by averaging 1000 sampled images. We choose 3 samples to display to illustrate the diversity of DIVNOISING results.

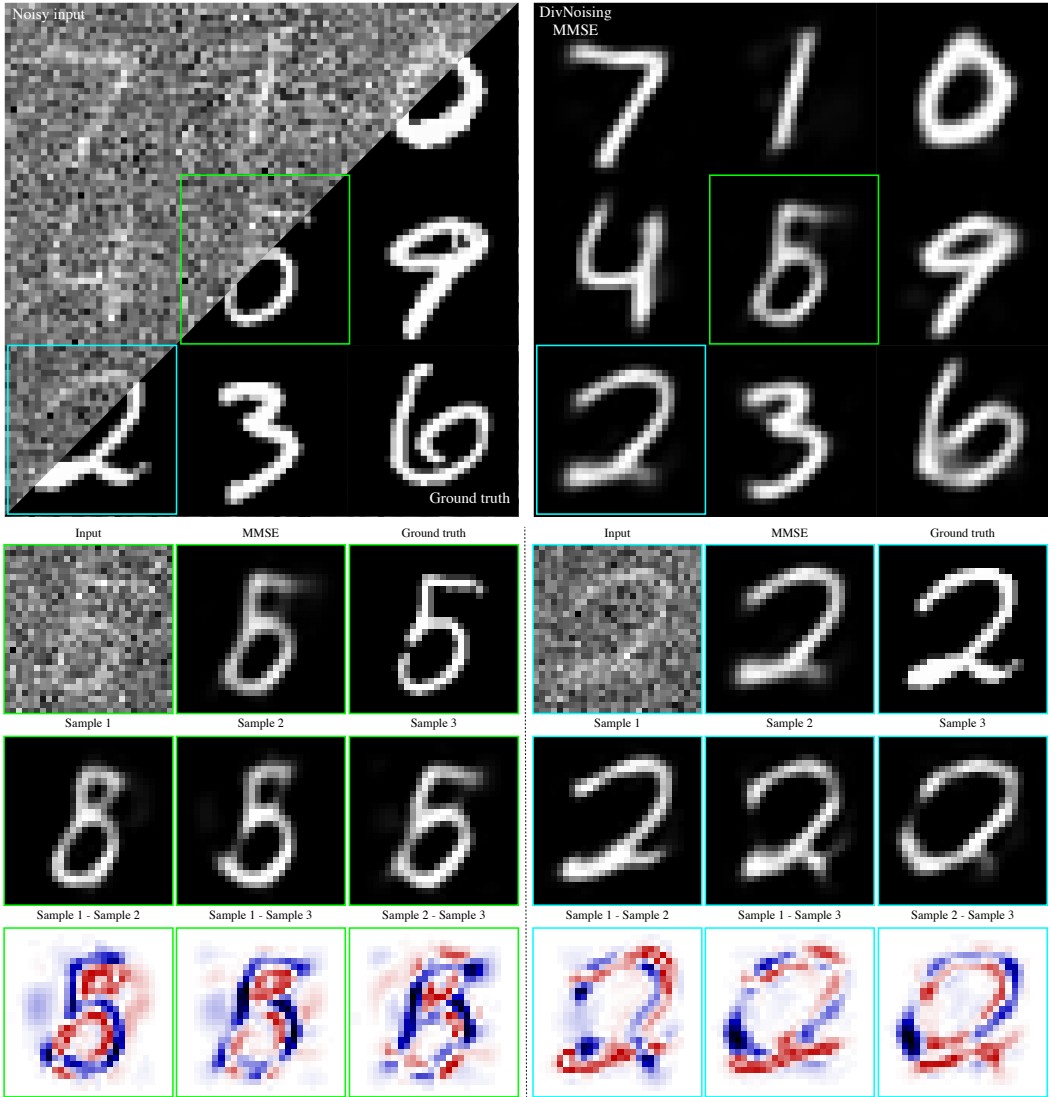

Figure 22: **Additional qualitative results for the *MNIST* LeCun et al. (1998) dataset.** Here, we show qualitative results for two cropped regions (green and cyan). The MMSE estimate was produced by averaging 1000 sampled images. We choose 3 samples to display to illustrate the diversity of DivNoising results.

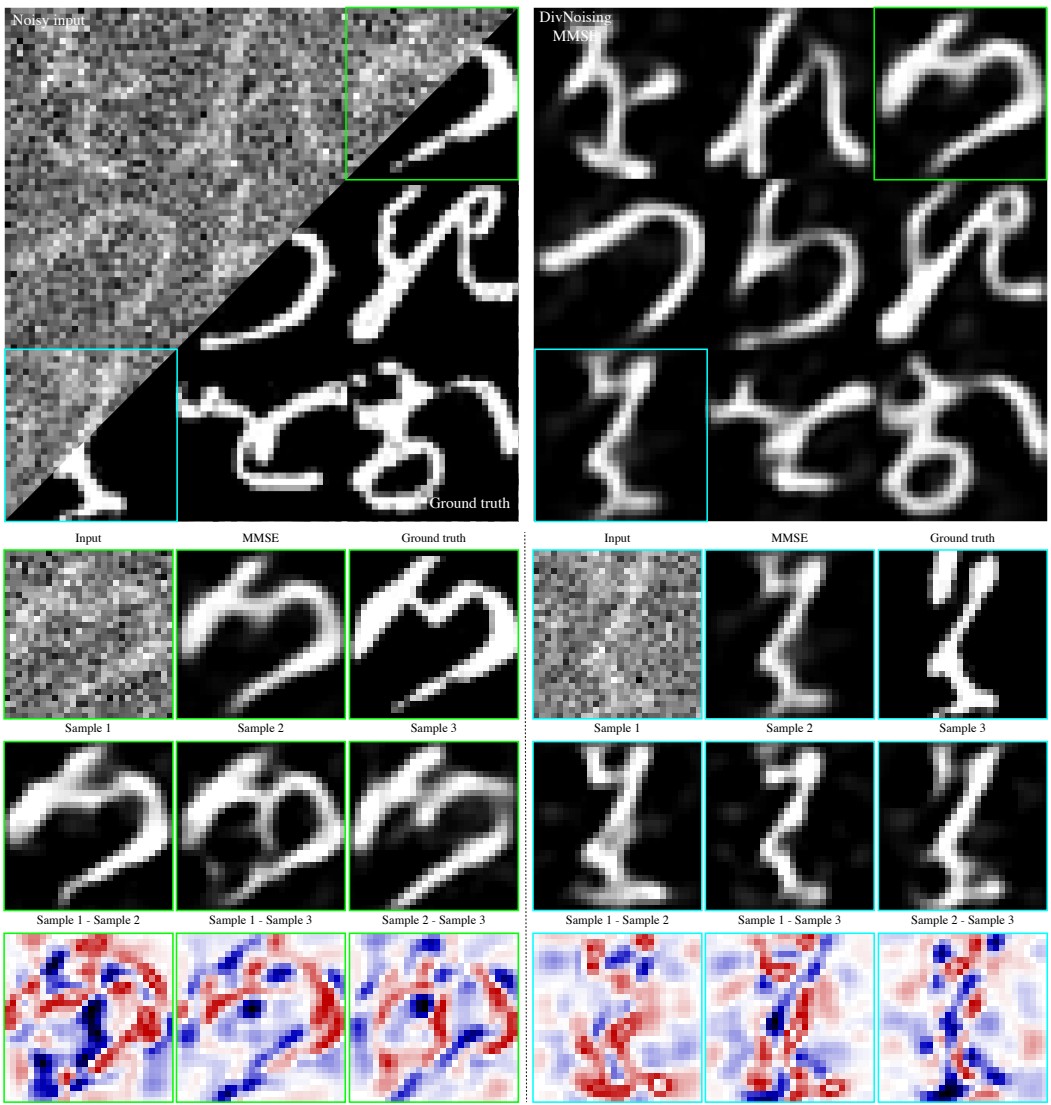

Figure 23: **Additional qualitative results for the *KMNIST* Clanuwat et al. (2018) dataset.** Here, we show qualitative results for two cropped regions (green and cyan). The MMSE estimate was produced by averaging 1000 sampled images. We choose 3 samples to display to illustrate the diversity of DIVNOISING results.

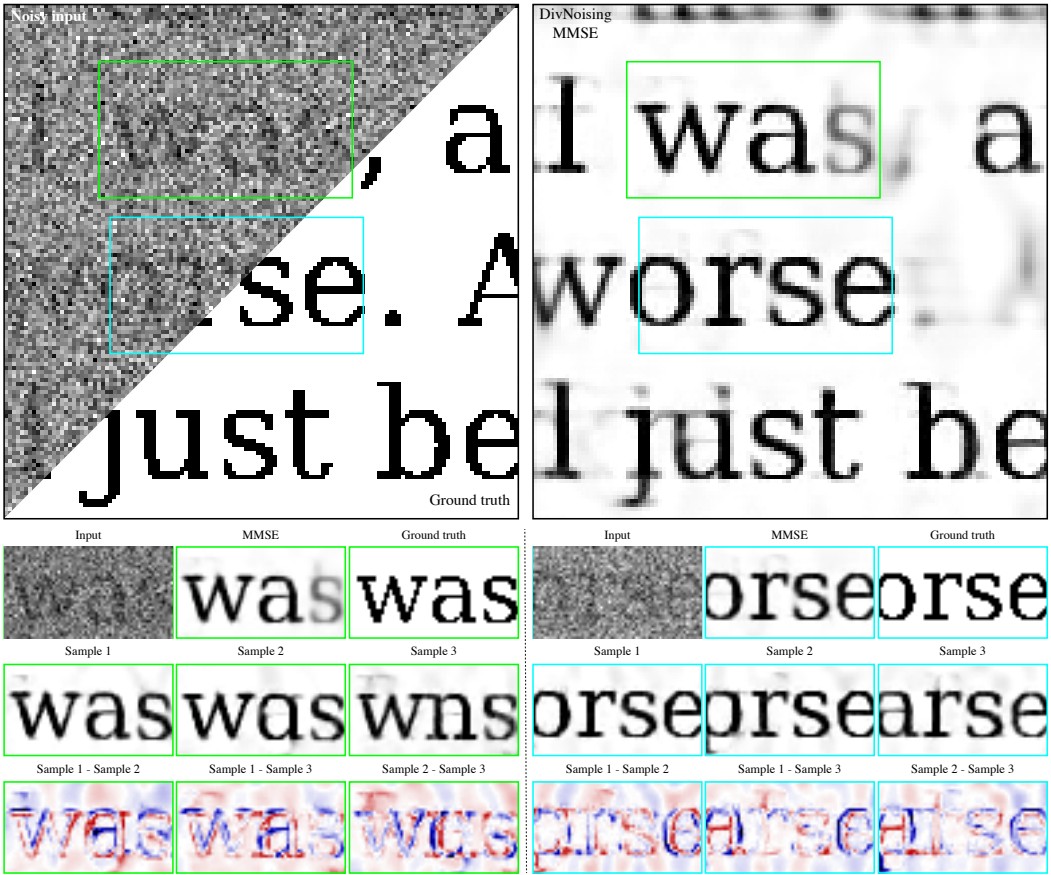

Figure 24: **Additional qualitative results for the *eBook* Marsh (2004) dataset.** Here, we show qualitative results for two cropped regions (green and cyan). The MMSE estimate was produced by averaging 1000 sampled images. We choose 3 samples to display to illustrate the diversity of DIVNOISING results.

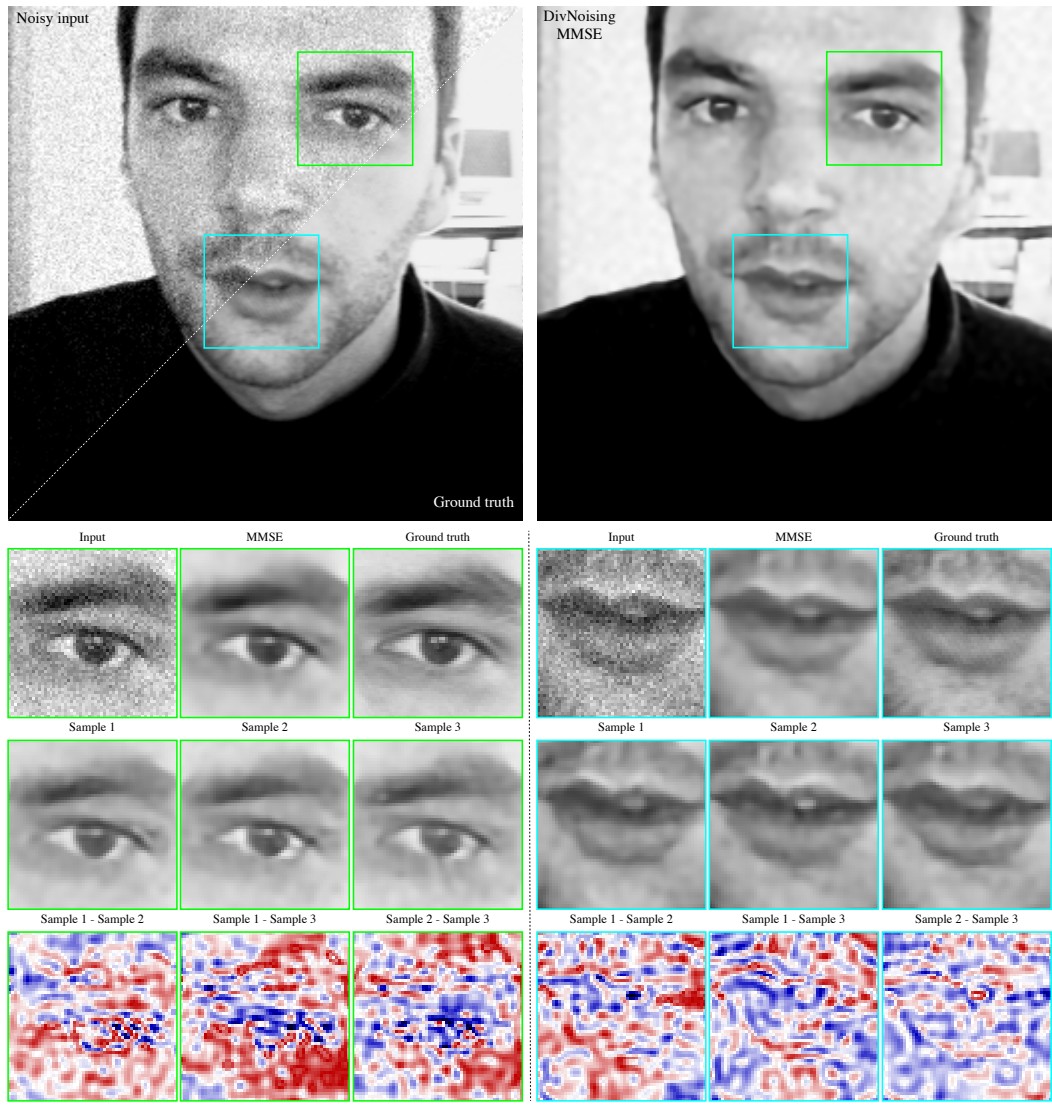

Figure 25: **Additional qualitative results for the *BioID Face* noa dataset.** Here, we show qualitative results for two cropped regions (green and cyan). The MMSE estimate was produced by averaging 1000 sampled images. We choose 3 samples to display to illustrate the diversity of DIVNOISING results.

## A.10 RESULTS ON NATURAL IMAGES

We investigated the denoising performance of DIVNOISING network on the natural images benchmark dataset $BSD68$ (Roth & Black, 2005) and show our results in Appendix Fig. 26, where the input has been corrupted with Gaussian noise of $\sigma = 25$. With our depth 2 network having 96 feature channels in the first network layer, we achieve a PSNR of $27.45$ dB while our unsupervised NOISE2VOID baseline gives 27.71 dB. As discussed in the main text, this does not come as a surprise since our DIVNOISING network is comparatively small and asked to learn a complete generative model of the entire data domain (see main text and Appendix Figs. 8-10). Learning such a model for the tremendous diversity present in natural images is challenging, and likely the reason why other architectures solving problems posed on the domain of natural images are much larger than our networks are. Future versions of DIVNOISING will address this issue by using more expressive architectures. However, DIVNOISING already gives us access to clean samples from the true (data) posterior (see Appendix Fig. 26).

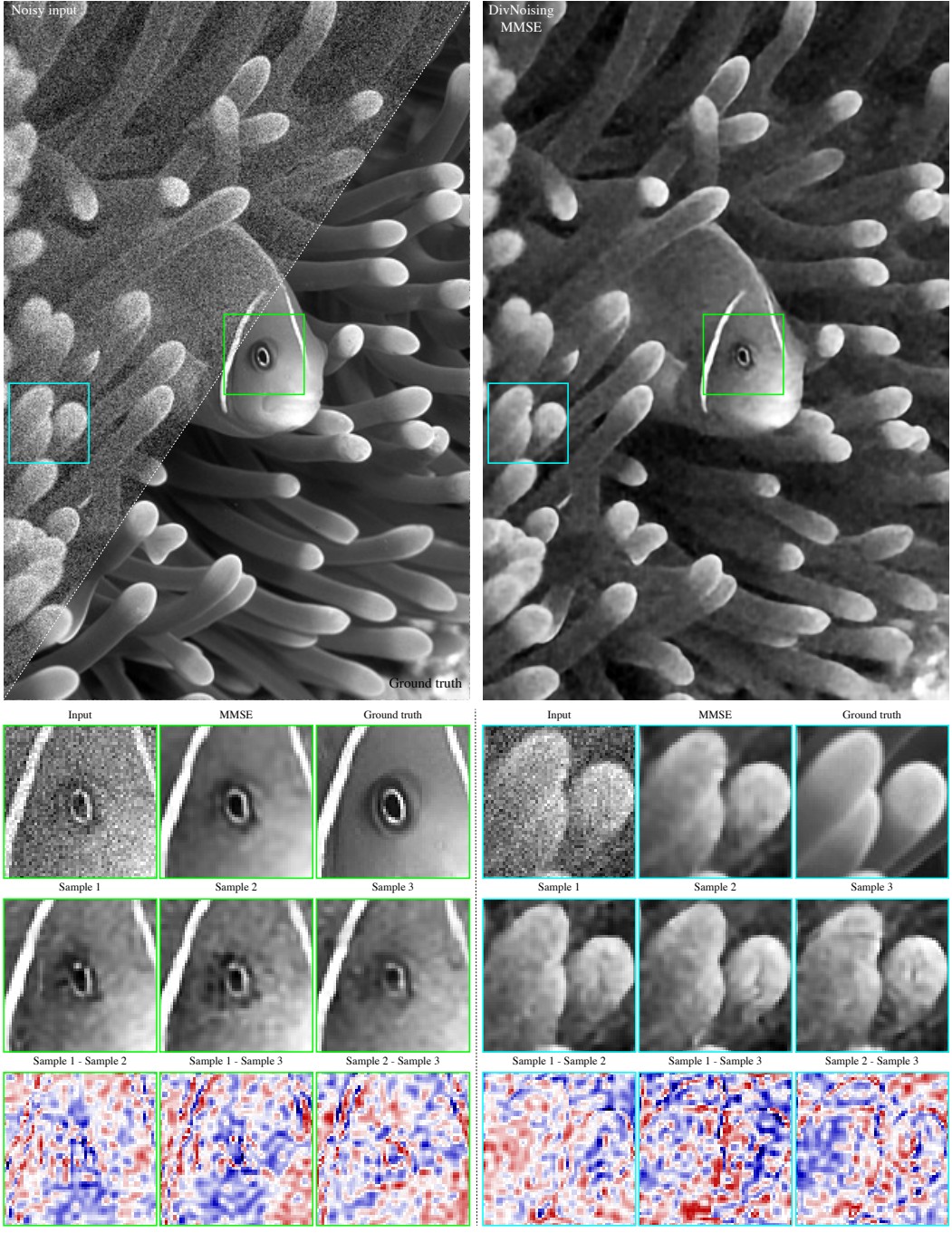

Figure 26: **Qualitative results for the *BSD68* dataset Roth & Black (2005).** Our relatively small DIVNOISING networks fail to capture the ample structural diversity present in natural photographic images thereby exhibiting sub-par performance. However, diversity at adequately small image scales (with respect to the used network's capabilities) can still be observed, as demonstrated with the different samples and the difference images corresponding to the green and cyan insets. We are confident that future work on DIVNOISING with larger networks and different network architectures/training schedules will expand the capabilities of this method to capture more complex image domains.

### A.11 How Accurate is the DivNoising Model and the Approximate Posterior?

Upon close inspection, we find that the images sampled by DivNoising exhibit various imperfections, making clear that they are in fact only samples from an approximate posterior.

For example, we find that DivNoising samples are often smoother than real images, see *e.g.* Appendix Figs. 15 and 22. We attribute this problem to our network architecture (see also Appendix Section A.9. For instance, a U-Net based supervised denoiser can make use of skip connections to propagate high frequency information. But DivNoising VAEs have to pipe all information through the downsampled latent variable bottleneck.

Another common artefact in sampled images is the presence of faint overlayed structures in the background (see Suppl. Fig. 24). Note that this artefact is less pronounced than in the MMSE estimate (where we expect such artefacts).

We believe that most of these remaining issues will be solved/reduced by using more sophisticated network architectures and refined training schedules.

### A.12 Derivation of DivNoising Loss Function from Probability Model Perspective

Here, we want to provide a more formal derivation of why our loss function can be used to train the VAE as desired. We follow a similar line of argument as has been laid out for the standard VAE by Doersch in (Doersch, 2016).

In our framework, we assume that the observed data $\mathbf{x}$ is generated from some underlying latent variable $\mathbf{z}$ through some clean signal $\mathbf{s}$ via a known noise model $p_{\mathrm{NM}}(\mathbf{x}|\mathbf{s})$. This process of data generation is depicted as a graphical model shown in Appendix Fig. 27.

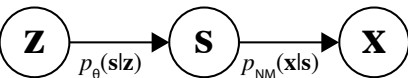

Figure 27: **Graphical model of the data generation process.**

The decoder describes a full joint model for all three variables:

$$p_\theta(\mathbf{z}, \mathbf{x}, \mathbf{s}) = p(\mathbf{x}, \mathbf{s}|\mathbf{z})p(\mathbf{z}) = p(\mathbf{x}|\mathbf{s}, \mathbf{z})p_\theta(\mathbf{s}|\mathbf{z})p(\mathbf{z}) \tag{6}$$

In the assumed graphical model in (Appendix Fig. 27) $\mathbf{x}$ is conditionally independent of $\mathbf{z}$ given $\mathbf{s}$. Formally, this implies that

$$p(\mathbf{x}|\mathbf{s}, \mathbf{z}) = p_{\mathrm{NM}}(\mathbf{x}|\mathbf{s}). \tag{7}$$

Using Supp. Eq. 7, we can reformulate Supp. Eq. 6 as

$$p_\theta(\mathbf{z}, \mathbf{x}, \mathbf{s}) = p_{\mathrm{NM}}(\mathbf{x}|\mathbf{s})p_\theta(\mathbf{s}|\mathbf{z})p(\mathbf{z}). \tag{8}$$

To train the generative model from Appendix Fig. 27 we try to adjust the parameters $\theta$ to maximize the likelihood of observing our training data $\mathbf{x}$. This means that we need to maximize

$$p_\theta(\mathbf{x}) = \int p_{\mathrm{NM}}(\mathbf{x}|\mathbf{s} = g_\theta(\mathbf{z}))p(\mathbf{z})d\mathbf{z}. \tag{9}$$

However, computing the integral in Supp. Eq. 9 is intractable due to the high dimensionality of $\mathbf{z}$. In our particular model, we would need to integrate over $64$ dimensions for each pixel for all our datasets except MNIST and KMNIST datasets where we would need to integrate over $8$ dimensions for each pixel. An alternative to computing the integral would be to approximate it by sampling a large number of values $\mathbf{z}^1, \mathbf{z}^2, ..., \mathbf{z}^K$ from $p(\mathbf{z})$ and computing $p_\theta(\mathbf{x}) \approx \frac{1}{K}\sum_{k=1}^{K} p_{\mathrm{NM}}(\mathbf{x}|\mathbf{s} = g_\theta(\mathbf{z}^k))$. However, since $p_{\mathrm{NM}}(\mathbf{x}|\mathbf{s} = g_\theta(\mathbf{z}^k))$ will be very close to $0$ for almost all $\mathbf{z}^k$, this would require $K$ to be a very large number for each image in our training set.

Following the idea introduced in (Kingma & Welling, 2014), we overcome this problem by instead using an encoder to describe an auxiliary distribution $q_\phi(\mathbf{z}|\mathbf{x})$. The encoder can take a noisy image $\mathbf{x}$ and yield a distribution over $\mathbf{z}$ values, which in turn are likely to produce $\mathbf{x}$ under the generative model. We want the encoder distribution $q_\phi(\mathbf{z}|\mathbf{x})$ to approximate the true underlying distribution $q_\phi(\mathbf{z}|\mathbf{x}) \approx p_\theta(\mathbf{z}|\mathbf{x})$, as it is implicitly described by our graphical model. From Bayes theorem, $p_\theta(\mathbf{z}|\mathbf{x})$ factorizes as

$$p_\theta(\mathbf{z}|\mathbf{x}) = \frac{p_\theta(\mathbf{x}|\mathbf{z})p(\mathbf{z})}{p_\theta(\mathbf{x})}. \tag{10}$$

The decoder in DIVNOISING setup is a deterministic function of $\mathbf{z}$, i.e., $g_\theta(\mathbf{z}) = \mathbf{s}$. Hence, we can reformulate Supp. Eq. 10 as

$$p_\theta(\mathbf{z}|\mathbf{x}) = \frac{p_{\text{NM}}(\mathbf{x}|\mathbf{s} = g_\theta(\mathbf{z}))p(\mathbf{z})}{p_\theta(\mathbf{x})}. \tag{11}$$

We can describe the quality of the encoder distribution, *i.e.* how well it approximates the true $p_\theta(\mathbf{z}|\mathbf{x})$ via the KL divergence

$$\mathbb{KL}\left(q_\phi(\mathbf{z}|\mathbf{x})||p_\theta(\mathbf{z}|\mathbf{x})\right) = -\int q_\phi(\mathbf{z}|\mathbf{x}) \log \frac{p_\theta(\mathbf{z}|\mathbf{x})}{q_\phi(\mathbf{z}|\mathbf{x})} d\mathbf{z}. \tag{12}$$

Substituting Supp. Eq. 11 in Supp. Eq. 12, we get

$$\begin{aligned}
\mathbb{KL}\left(q_\phi(\mathbf{z}|\mathbf{x})||p_\theta(\mathbf{z}|\mathbf{x})\right) &= -\int q_\phi(\mathbf{z}|\mathbf{x}) \log \frac{p_{\text{NM}}(\mathbf{x}|\mathbf{s} = g_\theta(\mathbf{z}))p(\mathbf{z})}{p_\theta(\mathbf{x})q_\phi(\mathbf{z}|\mathbf{x})} d\mathbf{z} \\
&= -\int q_\phi(\mathbf{z}|\mathbf{x}) [\log \frac{p_{\text{NM}}(\mathbf{x}|\mathbf{s} = g_\theta(\mathbf{z}))p(\mathbf{z})}{q_\phi(\mathbf{z}|\mathbf{x})} - \log p_\theta(\mathbf{x})] d\mathbf{z} \\
&= -\int q_\phi(\mathbf{z}|\mathbf{x}) \log \frac{p_{\text{NM}}(\mathbf{x}|\mathbf{s} = g_\theta(\mathbf{z}))p(\mathbf{z})}{q_\phi(\mathbf{z}|\mathbf{x})} d\mathbf{z} + \int q_\phi(\mathbf{z}|\mathbf{x}) \log p_\theta(\mathbf{x}) d\mathbf{z} \\
&= -\int q_\phi(\mathbf{z}|\mathbf{x}) \log \frac{p_{\text{NM}}(\mathbf{x}|\mathbf{s} = g_\theta(\mathbf{z}))p(\mathbf{z})}{q_\phi(\mathbf{z}|\mathbf{x})} d\mathbf{z} + \log p_\theta(\mathbf{x}) \int q_\phi(\mathbf{z}|\mathbf{x}) d\mathbf{z}.
\end{aligned}$$

Since $\int q_\phi(\mathbf{z}|\mathbf{x}) d\mathbf{z} = 1$, we get

$$\mathbb{KL}\left(q_\phi(\mathbf{z}|\mathbf{x})||p_\theta(\mathbf{z}|\mathbf{x})\right) = -\int q_\phi(\mathbf{z}|\mathbf{x}) \log \frac{p_{\text{NM}}(\mathbf{x}|\mathbf{s} = g_\theta(\mathbf{z}))p(\mathbf{z})}{q_\phi(\mathbf{z}|\mathbf{x})} d\mathbf{z} + \log p_\theta(\mathbf{x}).$$

This implies

$$\begin{aligned}
\log p_\theta(\mathbf{x}) &= \int q_\phi(\mathbf{z}|\mathbf{x}) \log \frac{p_{\text{NM}}(\mathbf{x}|\mathbf{s} = g_\theta(\mathbf{z}))p(\mathbf{z})}{q_\phi(\mathbf{z}|\mathbf{x})} d\mathbf{z} + \mathbb{KL}\left(q_\phi(\mathbf{z}|\mathbf{x})||p_\theta(\mathbf{z}|\mathbf{x})\right) \\
&= ELBO + \mathbb{KL}\left(q_\phi(\mathbf{z}|\mathbf{x})||p_\theta(\mathbf{z}|\mathbf{x})\right),
\end{aligned} \tag{13}$$

where $ELBO$ is the *Evidence Lower Bound* as also introduced in (Kingma & Welling, 2019) in the context of standard VAEs and here $ELBO = \int q_\phi(\mathbf{z}|\mathbf{x}) \log \frac{p_{\text{NM}}(\mathbf{x}|\mathbf{s} = g_\theta(\mathbf{z}))p(\mathbf{z})}{q_\phi(\mathbf{z}|\mathbf{x})} d\mathbf{z}$. Note that the KL divergence term in Supp. Eq. 13 is always greater than or equal to 0 and hence, $ELBO$ is a lower bound for $\log p_\theta(\mathbf{x})$, i.e., $\log p_\theta(\mathbf{x}) \geqslant ELBO$. It follows from Supp. Eq. 13 that

$$ELBO = \log p_\theta(\mathbf{x}) - \mathbb{KL}\left(q_\phi(\mathbf{z}|\mathbf{x})||p_\theta(\mathbf{z}|\mathbf{x})\right) \tag{14}$$

Supp. Eq. 14 implies that maximizing ELBO with respect to $\phi$ and $\theta$ maximizes $\log p_\theta(\mathbf{x})$ and minimizes $\mathbb{KL}\left(q_\phi(\mathbf{z}|\mathbf{x})||p_\theta(\mathbf{z}|\mathbf{x})\right)$, the goals we seek to achieve. Hence,

$$\max ELBO = \max \left( \int q_\phi(\mathbf{z}|\mathbf{x}) \log \frac{p_{\text{NM}}(\mathbf{x}|\mathbf{s} = g_\theta(\mathbf{z}))p(\mathbf{z})}{q_\phi(\mathbf{z}|\mathbf{x})} d\mathbf{z} \right)$$

$$= \max \left( \int q_\phi(\mathbf{z}|\mathbf{x}) \log p_{\text{NM}}(\mathbf{x}|\mathbf{s} = g_\theta(\mathbf{z}))d\mathbf{z} + \int q_\phi(\mathbf{z}|\mathbf{x}) \log \frac{p(\mathbf{z})}{q_\phi(\mathbf{z}|\mathbf{x})} d\mathbf{z} \right)$$

$$= \max \left( \int q_\phi(\mathbf{z}|\mathbf{x}) \log p_{\text{NM}}(\mathbf{x}|\mathbf{s} = g_\theta(\mathbf{z}))d\mathbf{z} - \mathbb{KL}\left(q_\phi(\mathbf{z}|\mathbf{x})||p(\mathbf{z})\right) \right)$$

$$= \max \left( \mathbb{E}_{q_\phi(\mathbf{z}|\mathbf{x})}[\log p_{\text{NM}}(\mathbf{x}|\mathbf{s} = g_\theta(\mathbf{z}))] - \mathbb{KL}\left(q_\phi(\mathbf{z}|\mathbf{x})||p(\mathbf{z})\right) \right).$$

Maximizing the $ELBO$ is equivalent to minimizing the negative $ELBO$, thus giving us the DIVNOISING loss function

$$\mathcal{L}_{\phi,\theta}(\mathbf{x}) = \min(\mathbb{E}_{q_\phi(\mathbf{z}|\mathbf{x})}[-\log p_{\text{NM}}(\mathbf{x}|\mathbf{s} = g_\theta(\mathbf{z}))] + \mathbb{KL}\left(q_\phi(\mathbf{z}|\mathbf{x})||p(\mathbf{z})\right)), \tag{15}$$

where the expected value is approximated in each iteration by drawing a single sample from $q_\phi(\mathbf{z}|\mathbf{x})$. Note that the first term in the summation in Supp. Eq. 15 is the same as described in Section 5 in the main text whereas the second term in the summation is the same as used in the standard VAE loss.

### A.13   COMPARISON OF PREDICTED VARIANCES BY VARIOUS METHODS

Unsupervised DIVNOISING and vanilla VAEs are both trained fully unsupervised, learning to predict per pixel noise models. Learning a good noise model is essential for good denoising performance as evident from Table 1. Here, we compare the noise models and variance maps predicted for two datasets by our unsupervised DIVNOISING and vanilla VAEs.

***BioID Face* dataset.** This dataset has been synthetically corrupted with Gaussian noise of $\mu = 0$ and $\sigma = 15$.

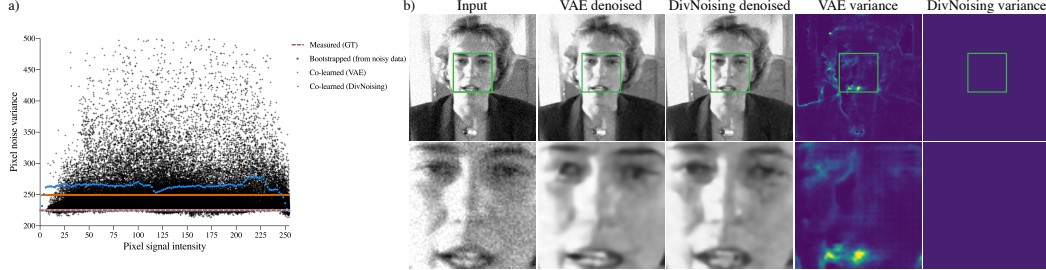

Figure 28: **Comparison of noise models and variance maps predicted by the vanilla VAE and DIVNOISING. (a)** For each predicted signal intensity (x-axis), we show the variance of noisy observations (y-axis). The plot is generated from experiments on the *BioID Face* dataset. The dashed red line shows the true noise distribution (Gaussian noise with $\sigma^2 = 225$). The noise model created via bootstrapping, and the noise model we co-learned with DIVNOISING, correctly show (approximately) constant values across all signal intensities. The implicitly learned noise model of the vanilla VAE has to independently predict the noise variance for each pixel. Its predictions clearly deviate from the true constant noise variance. **(b)** We visually compare the denoising results and show how the predicted variance varies across the image. While the variance predicted by the implicitly co-learned vanilla VAE model varies depending on the image content, the variance predicted by the co-learned DIVNOISING model correctly remains flat.

*Convallaria* **dataset.** This dataset is intrinsically noisy and the noise distribution resembles the shot noise and read out noise characteristics as typical for images acquired under low light settings.

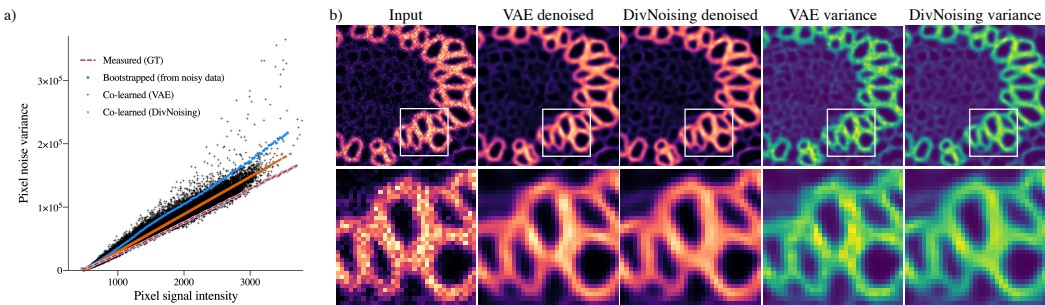

Figure 29: **Comparison of noise models and variance maps predicted by the vanilla VAE and DIVNOISING. (a)** For each predicted signal intensity (x-axis), we show the variance of noisy observations (y-axis). The plot is generated from experiments on the *Convallaria* dataset. The dashed red line shows the true noise distribution (measured from pairs of noisy and clean calibration data). We compare the noise model co-learned by the vanilla VAE and DIVNOISING with ground truth noise model and a noise model bootstrapped from noisy data alone as described in Prakash et al. (2020). Clearly, the noise model learnt by unsupervised DIVNOISING is a much better approximation to the ground truth noise model compared to the noise models learned/obtained by other methods. **(b)** We visually compare the denoising results and show how the predicted variance varies across the image. As a consequence of the approximately linear relationship between signal and noise variance, both variance images closely resemble the denoised results. However, the result of the vanilla VAE additionally contains artifacts.

## A.14   QUANTITATIVE COMPARISON OF DIVNOISING WITH SELF2SELF

Since SELF2SELF is trained per image, leading to prohibitive computation times on our test sets, we randomly chose single images for four of our datasets (*FU-PN2V Convallaria*, *FU-PN2V Mouse actin*, *FU-PN2V Mouse nuclei* and *W2S Ch.1 (avg1)*) which contain real-world noise.

We compare the performance of SELF2SELF trained on single images with the performance of DIVNOISING when trained on (*i*) the same single image as SELF2SELF, and (*ii*) the entire body of available noisy data in the respective dataset. All trained networks are then applied to the selected single images. Note that SELF2SELF is run with its default settings.

Since SELF2SELF training is computationally expensive even for a single image, we decided to limit training time to 10 hours per input on a NVIDIA TITAN Xp GPU. We monitored its performance by periodically computing the PSNR (every 3000 training steps), showing that even after 10 hours, SELF2SELF is not yet fully converged. Table 2 shows all results we obtained. It can be seen that DIVNOISING, when trained on the full dataset, leads consistently to better performance, while DIVNOISING trained on single images leads to comparable results in a fraction of training time and using significantly less GPU memory.

| Datasets | PSNR (dB) | | | Run time (hours) | | | GPU memory (GB) | | |
|---|---|---|---|---|---|---|---|---|---|
| | S2S | DivN.[1] | DivN.[all] | S2S | DivN.[1] | DivN.[all] | S2S | DivN.[1] | DivN.[all] |
| Convallaria | 36.23 | 36.42 | **36.94** | 10 | **0.44** | 10 | 11 | **1.5** | **1.5** |
| Mouse actin | 33.15 | 33.80 | **33.99** | 10 | **1.09** | 10 | 11 | **1.5** | **1.5** |
| Mouse nuclei | 36.21 | 35.99 | **36.46** | 10 | **0.16** | 7 | 11 | **1.5** | **1.5** |
| Ch.0 (avg1) | 31.40 | **31.81** | 31.59 | 10 | **0.48** | 3.75 | 11 | **1.5** | **1.5** |

Table 2: **Comparison of Self2Self with DivNoising.** We train Self2Self (S2S) on a random single image per dataset and compare it with DIVNOISING trained on the same single image (DivN.[1]) and DIVNOISING trained with the full dataset (DivN.[all]). All methods are tested on the selected single image. Overall best method is indicated in bold. For all datasets, DIVNOISING leads to best performance while being orders of magnitude faster and needing significantly less GPU memory.

## A.15 SAMPLING TIME DURING PREDICTION

During prediction, in order to obtain diverse results, or to compute the MMSE or MAP estimates, we need to sample multiple denoised images from the trained DIVNOISING posterior. Table 3 reports the time (in seconds) needed for sampling 1000 denoised images. For all datasets holds that sampling 1000 denoised images requires less than 7 seconds.

| | Datasets | time (sec) |
|---|---|---|
| FU-PN2V | Convallaria | $3.37 \pm 0.059$ sec |
| | Mouse Act. | $6.40 \pm 0.052$ sec |
| | Mouse Nuc. | $1.91 \pm 0.043$ sec |
| W2S | Ch.0 (avg1) | $3.42 \pm 0.074$ sec |
| | Ch.1 (avg1) | $3.37 \pm 0.068$ sec |
| | Ch.2 (avg1) | $3.38 \pm 0.053$ sec |
| | Ch.0 (avg16) | $3.39 \pm 0.060$ sec |
| | Ch.1 (avg16) | $3.40 \pm 0.062$ sec |
| | Ch.2 (avg16) | $3.40 \pm 0.048$ sec |
| DenoiSeg | Mouse | $1.17 \pm 0.034$ sec |
| | Flywing | $3.91 \pm 0.034$ sec |
| | Mouse s&p | $2.82 \pm 0.070$ sec |
| | BioID Face | $2.09 \pm 0.046$ sec |

Table 3: **Sampling times with DIVNOISING.** Average time ($\pm$ SD) needed to sample 1000 denoised images from a trained DIVNOISING network (evaluated over all test images of the respective dataset).

