# OpenReview forum: "Fully Unsupervised Diversity Denoising with Convolutional Variational Autoencoders"
_ICLR.cc/2021/Conference — ICLR 2021 Poster_

### Official Review · AnonReviewer3 · 2020-10-19
**The idea sounds reasonable. But the novelty seems marginal and I have several concerns about the application range of the proposed method.**

**Rating:** 6
**Confidence:** 3

**Review:**

This paper proposes a modified VAE for unsupervised image denoising. Unlike existing methods only predicting a single denoised image, the proposed one can generate diverse and plausible results. The experimental results show that this method can outperform existing unsupervised denoising methods. I have several concerns about this paper.

(1) The modifications relative to Kingma & Welling (2014) are marginal. The proposed method more likes an application of Kingma & Welling (2014).

(2) According to Eq. 1, the noises are pixel independent which does not hold for most of the applications e.g. images after demosaicing or ISP denoising. How can the proposed method deal with it?

(3) VAE framework seems only to work for small and constrained images. I am wondering about the denoising results of the proposed method for high-resolution nature images?

(4) At the beginning of Sec. 5, the authors claim that the noise model should be known. How does the proposed method deal with the noises without the noise model for example images after demosaicing or ISP denoising?
-------------------------------------------------------------------------------------------------------------------------------------

I appreciate that this paper has some merits. But I lower my rating because of the limited usage of the proposed method.
It seems it cannot handle high-resolution nature images. Most of the experiments use constrained images e.g. biomedical images or images with small resolution.

---

> ### Author Response · Authors · 2020-11-12
> **Initial Point by Point Rebuttal (and invitation for further discussion)**
>
> We thank the reviewer for her/his insightful comments and overall encouraging evaluation of our work. We address the concerns raised point by point below.
>
> **Remark 1: Comparison to Kingma & Welling (2014)**
> It is true that our method relies on the VAE framework by Kingma and Welling (2014). Our novel contribution is the formulation of the unsupervised denoising problem within this framework, by including the imaging noise distribution (noise model) in the decoder. This formulation has, to our knowledge, never been attempted before.
> While it is possible to achieve a denoising effect by directly applying Kingma’s VAEs without an explicit noise model in the decoder, the original VAE was not designed for denoising and has to effectively predict a separate noise model for each pixel.
> We clearly highlight our advantages over this approach in the subsection “Denoising with vanilla VAEs” as well as in Fig. 3, Table 1, and Appendix Figures 27 and 28.
>
> Additionally, we propose a new fully unsupervised DivNoising setup as well, where the decoder can learn the noise model on the fly as a function of the predicted signal.
> To summarize, our contributions are threefold:
> *(i)* We are the first to present an unsupervised denoising method enabling the sampling of diverse solutions from an approximated true signal posterior distribution (in contrast to state-of-the-art baselines as well as Quan et al (2020) referenced by Reviewer 4).
> *(ii)* We propose a fully unsupervised DivNoising variant that does not require an apriori known noise model.
> *(iii)* Both of our proposed variants allow us to learn an approximation to the real posterior, from which we can then sample efficiently. These individual samples are useful for downstream processing as shown with applications to OCR and cell segmentation. Additionally, we show how to get point estimates such as MMSE and MAP (which is not obtainable with other methods).
>
> **Remarks 2 and 4: How can the proposed method deal with images after demosaicing or ISP denoising?**
> In the current form, our method does not deal with such tasks. In this paper we are specifically looking at pixel-noises, which are the dominating sources of noise in biomedical microscopy images. (Note that the domains we focus on are of extraordinarily high practical relevance and that many biomedical research projects will directly benefit from employing DivNoising in their downstream analyses.)
> However, if future work describes how suitable noise models $p(x|s)$ can be constructed/learned for other/more complex types of noise, then our approach will be able to use them. We want to emphasize that our contribution is to demonstrate a principled way to use noise models in VAEs for diverse and unsupervised denoising, an application/direction that was never explored before.
>
> **Remark 3: Denoising results of the proposed method for high resolution nature images**
> We discuss the results for natural images on benchmark BSD68 dataset in our discussion section and also show results of our method on this dataset in Appendix Fig. 25. More discussion about the same is found in Appendix Section A10.
> We find that for such images, we perform almost on par with our baseline Noise2Void, but we additionally learn the full posterior which gives us access to diverse sampling unlike other methods. Not outperforming baseline methods on diverse natural images is in line with our expectations since natural images have tremendous content diversity and learning a full generative model of such data will require networks with higher capacity. (Note that virtually all image generation methods and applications focus on rather narrow domains such as faces or other limited classes of objects.) However, we strongly believe that with more sophisticated VAE architectures, DivNoising will improve also on rich domains such as natural images.
> Still, we want to remind the reviewer that the image domain we target in our work is of extraordinarily high value for many researchers that analyze microscopy images on a daily basis as well as for OCR and facial recognition applications in forensics which often rely on image restoration.
>
> **Finally**, we want to encourage the reviewer to further comment and discuss our responses. We are very open to further constructive criticism and are happy to include clarifications in the final version of paper. Thanks! :)

---

> > ### Comment · AnonReviewer3 · 2020-11-19
> > **concerns**
> >
> > I admit that the proposed framework is useful in some scenarios and interesting. However, if it can only handle low-resolution images with known pixel independent noises, the usage is quite limited. In the rebuttal, the authors claim that they test their method in BSD. But the resolution of it is somewhat small (less than 500). There are many other datasets with high-resolution (e.g. Urban100, DIV2K). Also, the authors claim that 'if future work describes how suitable noise models can be constructed/learned for other/more complex types of noise, then our approach will be able to use them.' I do not think it is easy to model the real noises. Furthermore, if the noise model is known, we can directly generate noisy and clean image pairs for supervised learning with better performance.

---

> > > ### Author Response · Authors · 2020-11-19
> > > **Resolution concern is a possible misunderstanding; noise model concern is only valid in the natural image domain - our paper has a strong focus on biomedical image data though...**
> > >
> > > In the context of natural images we agree with most of the reviewer's comments but not in the context of data domain we target. Regardless of the domain, our approach already handles high resolution images (1024x1024; explanation follows below as minor comment). But first we address the question of noise models in the context of our target data domain.
> > >
> > > Our paper is clearly targeted for the important domain of **biomedical images (microscopy images)** in the context of computational biology, an officially listed subject area of ICLR (https://iclr.cc/Conferences/2021/CallForPapers). In virtually all practical setups **clean (pixel-noise free) data can simply not be acquired**, thereby invalidating the idea of using the noise models to create paired data for supervised training (which would require access to the unobtainable clean data). But **even if some clean data would be available, a correct noise model is usually not**. Also, this is consequently invalidating the idea of image pair generation for supervised training. Hence, the **ability of DivNoising to bootstrap or co-learn a suitable noise model from noisy images alone** is another **key contribution** of our work.
> > >
> > > Additionally, in microscopy image data, **pixel-wise noise is the most dominant "real noise”** (see [1-6]), rendering our method very practical and useful for many analysis tasks in computational biology.
> > >
> > > Minor comments:
> > > * While BSD68 images have indeed a limited pixel-resolution, our Convallaria data in the paper, for example, is 1024x1024, but due to the fully convolutional nature of DivNoising this is by no means the upper limit. Hence, our method is not limited by resolution of images.
> > > * Our work does not focus on denoising of natural images. BSD68 is a very standard natural image denoising dataset and has been used by numerous state-of-the-art methods for benchmarking (see [7-10]). DivNoising results for this dataset was included in the appendix to have a reference to other existing methods in the context of natural images. It is not central to our work. While we could use Urban100 or DIV2K data as well, these are not (yet) commonly used benchmarking denoising datasets. In general, we would prefer not to put additional emphasis on results on natural images. Still, if strongly requested, we can run DivNoising on Urban100 and add results to the appendix in the final version.
> > >
> > > In summary, while DivNoising certainly has merit in multiple domains, **we would like to be evaluated in the context of biomedical data and its specific limitations**.
> > >
> > > **References** _(sorry for terrible formatting, but the markdown parser of OpenReview is very limiting):_
> > >
> > > [1] Weigert, Martin, et al. "Content-aware image restoration: pushing the limits of fluorescence microscopy." Nature methods 15.12 (2018): 1090-1097.
> > >
> > > [2] Krull, Alexander, et al.. "Probabilistic noise2void: Unsupervised content-aware denoising." arXiv preprint arXiv:1906.00651 (2019).
> > >
> > > [3] Prakash, Mangal, et al. "Fully unsupervised probabilistic noise2void." 2020 IEEE 17th International Symposium on Biomedical Imaging (ISBI). IEEE, 2020.
> > >
> > > [4] Khademi, Wesley, et al. "Self-Supervised Poisson-Gaussian Denoising." arXiv preprint arXiv:2002.09558 (2020).
> > >
> > > [5] Zhou, Ruofan, et al. "W2S: A Joint Denoising and Super-Resolution Dataset." arXiv preprint arXiv:2003.05961 (2020).
> > >
> > > [6] Zhang, Yide, et al. “A poisson-gaussian denoising dataset with real fluorescence microscopy images." Proceedings of the IEEE Conference on Computer Vision and Pattern Recognition. 2019.”
> > >
> > > [7] Zhang, Kai et al., "Beyond a Gaussian Denoiser: Residual Learning of Deep CNN for Image Denoising."arXiv preprint arXiv:1608.03981 (2016).
> > >
> > > [8] Lehtinen, Jaakko, et al. "Noise2noise: Learning image restoration without clean data." arXiv preprint arXiv:1803.04189 (2018).
> > >
> > > [9] Krull, Alexander, et al. "Noise2void-learning denoising from single noisy images." Proceedings of the IEEE Conference on Computer Vision and Pattern Recognition. 2019.
> > >
> > > [10] Quan, Yuhui, et al. "Self2Self With Dropout: Learning Self-Supervised Denoising From Single Image." Proceedings of the IEEE/CVF Conference on Computer Vision and Pattern Recognition. 2020.

---

### Official Review · AnonReviewer2 · 2020-10-20
**Discussion about p(s) is preferred**

**Rating:** 7
**Confidence:** 3

**Review:**

This paper proposes a new method of noise removal using convolutional VAE.  An observed image with noise is input to VAE, and after the expression $z$ in the latent space, the noise removed image is finally output.  After that, it is possible to generate a pseudo noisy observation image according to the noise model.  The noise model part is flexibly designed using the Gaussian mixture model.  In the training, VAE and noise model can be learned at the same time. Since VAE is a generative model, and a clean denoising image can be obtained by averaging a large number of candidates of clean images, $s$, sampled from the periphery of the latent space representation $z$.

The advantages of the proposed method are that it is Fully unsupervised and that the noise model can be learned at the same time.

The paper itself is interesting, and the proposed method itself is good as one of the image processing methods, but there is a lack in the explanation part.  In this paper, the noise model $p (x|s)$ is well discussed, but I think that the discussion about the image prior, $p(s)$, for clean image is weak. As explained in p3 of the paper, $p(s|x) \propto p (x | s) p (s)$ is an important formula. In conventional image processing, $p(s)$ has been interpreted as smoothness or non-local similarity. In this method, it is treated as $p (s | z) p (z)$, but I think we should discuss a little more about what this is.

In the proposed method, there is no special assumption about the prior knowledge of the image, and it seems that it is normally distributed in the latent space. In this case, a large amount of data or some help of a network structure is needed to remove noise. The use of the convolutional structure in the proposed method is an indispensable element in the explanation of $p(s)$.

The noise reduction effect of the convolutional structure itself is well known after the study of Deep Image Prior, and recent research [a, b] also discusses the relationship between convolutional neural networks and non-local similarity. These may help the discussion about $p(s)$.

[a] Yokota, Tatsuya, et al. "Manifold Modeling in Embedded Space: A Perspective for Interpreting" Deep Image Prior "." ArXiv preprint arXiv: 1908.02995 (2019).
[b] Tachella, Julián, Junqi Tang, and Mike Davies. "CNN Denoisers As Non-Local Filters: The Neural Tangent Denoiser." ArXiv preprint arXiv: 2006.02379 (2020).

-------------
As a result of the feedback, the part I was concerned  became clear, so I would like to raise the score.

---

> ### Author Response · Authors · 2020-11-12
> **Initial Point by Point Rebuttal (and invitation for further discussion)**
>
> We thank the reviewer for an encouraging evaluation of our paper and we very much appreciate the insightful feedback.
>
> We agree that the prior over clean images $p(s)$ deserves more discussion.
> Here, we **want to attempt a clarification, which we suggest could in a similar form be included in the final version of the paper**.
>
> As you mention in your review, traditional methods often explicitly model $p(s)$ e.g. as a function of smoothness, presuming a higher apriori probability for smooth images.
> Albeit leading to respectable results, this is obviously an oversimplification. We can expect the true distribution $p(s)$ of clean images to be much more complex (e.g. for a particular experimental setup in a fluorescence microscope).
> Instead of explicitly modelling $p(s)$, our VAE based approach only implicitly describes $p(s)$ as $\int p_\theta(s|z) p(z) dz$ over all possible values of $z$.
> While the prior $p(z)$ is indeed assumed to be a unit Gaussian distribution, the conditional distribution $p_{\theta}(s|z)$ is learned by the decoder network.
> It is a dirac distribution centered at $g_{\theta}(z)$, i.e. the function implemented by the decoder network.
> Depending on its parameters, the network will implement the function differently, which ultimately leads to a different $p(s)$.
> Unlike with the traditional smoothness prior, we cannot directly calculate a probability density for $p(s)$ for a given $s$.
> However, to check the plausibility of the learned distribution, we can generate samples from $p(s)$ by first sampling from the unit Gaussian $p(z)$ in latent space and then feeding the samples through the decoder network. We show this in Figures 8-10 in our appendix.
> Albeit not perfect, we believe that our implicit prior produces plausible results, especially when considering local structures.
>
> As you point out in your review, since the work on ‘deep image prior’, convolutional neural networks are understood to have an inherent denoising effect.
> We believe this makes them especially suitable for our task and contributes to the quality of our results and the quality of the learned $p(s)$.
> However, note that the denoising quality we achieve is more than the sheer result of the convolutional architecture, which can for example be seen by the fact that we improve over all other convolutional baselines.
> Instead, to separate image content and noise, we rely on a generative model of image generation, which includes an appropriate noise model. We argue in Appendix Section A7 and show in Figures 11 and 12 in our appendix that deviating from the correct noise model (by changing $\beta$) worsens the results. (Also note that DivNoising outperforms results obtained with a regular VAE that does not employ a suitable noise model, as we show in Table 1 and  in Figures 27 and 28 in our appendix.)
>
> Last but not least, **we are happy to include and discuss the mentioned literature in the updated version of our paper** alongside the already cited Deep Image Prior.
>
> **Finally**, we want to encourage the reviewer to further comment and discuss our responses. We are very open to further constructive criticism and are happy to include clarifications in the final version of paper.
> Thanks! :)

---

> > ### Comment · AnonReviewer2 · 2020-11-17
> > **Thank you for your feedback**
> >
> > Thank you for adding the discussion on prior distribution, $p(s)$. I generally agree with the problem that methods defining $p(s)$ explicitly often over-modify natural images. On the other hand, improvements in accuracy have been reported by methods that do not explicitly define $p(s)$, such as deep image prior and the proposed method here.  While these approaches offer the benefits of improved accuracy, I feel they are black-box-like, and I find it difficult to interpret. I hope that by adding this discussion, the black box-like part will become clearer.  The literature [a,b] is some work trying be the clear the black-box in deep image prior and convolutional neural networks.
> >
> > In my subjectivity, this paper is very interesting and I support it.

---

### Official Review · AnonReviewer1 · 2020-10-27
**Official Blind Review #1**

**Rating:** 7
**Confidence:** 4

**Review:**

This paper devises a novel unsupervised denoising paradigm, DIVNOISING, that allows us, for the first time, to generate diverse and plausible denoising solutions, sampled from a learned posterior. This approach only requires noisy images and a suitable description of the imaging noise distribution, providing a new perspective for the image denoising field. It has demonstrated that the quality of denoised images is highly competitive, typically outperforming the unsupervised state-of-the-art, and at times even improving on supervised results. This paper is well-written and good-organized. However, the reviewer has the following concerns.

Q1. In DIVNOISING, noisy images are been created from a clean signal $s$ via a known noise model, i.e., $x\sim p_{NM}(x|s)$. One main practical concern is that is there any assumptions about this noise model and which type of noises are mainly considered to be removed?

Q2. The authors have demonstrated the excellent performance of DIVNOISING on several datasets, especially on microscopy datasets. Whether this denoiser can be applied to other noisy scenarios, such as a real color noisy image or MRI.

Q3. Open the source codes.

---

> ### Author Response · Authors · 2020-11-12
> **Initial Point by Point Rebuttal (and invitation for further discussion)**
>
> We thank you for your encouraging comments on our approach. We also thank you for your insightful questions. We address these questions one by one below:
>
> **Remark:** “is there any assumptions about this noise model and which type of noises are mainly considered to be removed?”
> The scope of this paper is to deal with per pixel noise as it commonly appears in microscopy data.  Such noise is inflicted by limited illumination and detector/camera imperfections and is the dominant source of noise in biomedical imaging [1,2,3,4,5], astronomy [6], as well as raw photography data [7,8].
> This being said, our approach can in principle be applied to any image as long as a probabilistic noise model can be described. We show, for the types of noise we consider (i.e. Poisson shot noise, Gaussian readout noise, and salt&pepper noise), how appropriate noise models can be measured, bootstrapped, or co-learned (fully unsupervised).
>
> **Remark:** “Whether this denoiser can be applied to other noisy scenarios, such as a real color noisy image or MRI.”
> In the context of microscopy image denoising an extension to RGB images is not needed. Still, such an extension is simple engineering and we will eventually include this into our open source framework.
> Other biomedical imaging modalities, such as MRI or CT (tomography) are studied by many groups, and we are ourselves looking into extension of the ideas presented here in order to restore said imagery. The challenge is to define and learn suitable noise models that go beyond per-pixel noise and can capture larger spatial dependencies.
> The same is true for removing noise from consumer level RGB images that have been subject to compression or demosaicing.
>
> **Remark:** “Open the source codes.”
> We did not disclose the source code for submission for reasons of anonymity. Our lab is proud to say that we provide the sources for all our methods and put extra emphasis on reproducibility of all reported results. (For an anonymized preview during the rebuttal period, please request in a comment below.)
>
> **Finally**, we want to encourage the reviewer to further comment and discuss our responses. We are very open to further constructive criticism and are happy to include clarifications in the final version of paper.
> Thanks! :)
>
> [1]  Weigert, Martin, et al. "Content-aware image restoration: pushing the limits of fluorescence microscopy." Nature methods 15.12 (2018): 1090-1097.
>
> [2] Krull, Alexander, et al.. "Probabilistic noise2void: Unsupervised content-aware denoising." arXiv preprint arXiv:1906.00651 (2019).
>
> [3] Prakash, Mangal, et al. "Fully unsupervised probabilistic noise2void." 2020 IEEE 17th International Symposium on Biomedical Imaging (ISBI). IEEE, 2020.
>
> [4] Khademi, Wesley, et al. "Self-Supervised Poisson-Gaussian Denoising." arXiv preprint arXiv:2002.09558 (2020).
>
> [5] Zhou, Ruofan, et al. "W2S: A Joint Denoising and Super-Resolution Dataset." arXiv preprint arXiv:2003.05961 (2020).
>
> [6] Beckouche, Simon et al. "Astronomical image denoising using dictionary learning." Astronomy & Astrophysics 556 (2013): A132.
>
> [7] Kumar, Prashanth et al. "Low Rank Poisson Denoising (LRPD): A Low Rank Approach Using Split Bregman Algorithm for Poisson Noise Removal From Images." CVPR Workshops. 2019.
>
> [8] Laine, Samuli, et al. "High-quality self-supervised deep image denoising." Advances in Neural Information Processing Systems. 2019.

---

### Official Review · AnonReviewer4 · 2020-10-28
**Estimating a distribution of denoised images instead of a single denoised image is not new (ref missing), but an interesting alternative was proposed. I am concerned on evaluations.**

**Rating:** 6
**Confidence:** 5

**Review:**

This manuscript proposed an interesting method to infer the distribution of denoised images for a given input instead of a single denoised image. Then, the final denoise image was generated using MMSE (averaging multiple denoised images) or using MAP (finding the mode of the posterior distribution).

1) Unfortunately, this manuscript missed one very important relevant recent work:
[Quan2020] Quan et al., Self2Self With Dropout: Learning Self-Supervised Denoising From Single Image, CVPR 2020.
This work also infers the distribution of denoised images instead of a single denoised image and yielded state-of-the-art results for both synthetic noise, real noise, salt-and-pepper noise and inpainting.
The ways of implementing distribution generations seem similar ("a principled approach to incorporate explicit models of the imaging noise distribution in the decoder of a VAE" in this manuscript and also similar structure was implemented in the decoder part of the network in [Quan2020]) except that the proposed method here utilized random sampling in the latent space while [Quan2020] used dropout to implement it. In my view, [Quan2020] and the proposed method share some common aspects that can undermine an important contribution of this manuscript in its current form, and thus this manuscript must explain the advantages and novelty of the proposed method over [Quan2020] properly.

Even though the proposed method with MAP is still new, for now [Quan2020] seems to have advantages over the proposed method such as (a) Self2Self requires a single noisy image while the proposed method requires noisy dataset, thus Self2Self can cover the case in this manuscript while the proposed method may not be able to cover the case of [Quan2020] with a single noisy image, (b) Self2Self yielded state-of-the-art performance on well known datasets such as Set9, BSD68, thus the readers can know that Self2Self is one of the state-of-the-art denoising methods. Even though the proposed method was evaluated on 13 datasets, none of them are used to evaluate denoisers in general. Therefore, it is hard to justify that the current proposed method also achieved state-of-the-art performance. (c) Self2Self was evaluated on both color, gray-scale images with synthetic noises as well as real noise (PolyU) while the proposed method was evaluated only on gray-scale images with real noise.

In addition, the way of implementing the generation network for samples was already proposed in [Kohl2018] for segmentation task. Thus, the contribution of the proposed method could be weakened substantially according to [Kohl2018], too, even though this manuscript nicely demonstrated that this sampling method in [Kohl2018] worked well for denoising problems.

2) Table 1 shows 13 datasets, but it is hard to use them to compare with other denoiser results in other literature. Many denoising works are using common datasets such as Set9, BSD68 (for synthetic noise), and often PolyU (for real noise) and it is helpful to have these results (at least in the appendix) for the comparison purpose with other previous works. Moreover, most images that were used in this manuscript seem to have specific structures (cells, characters), so I am not convinced that the proposed method will be generalized to regular images and color images. The proposed method has noise estimation components (co-learned, bootstrapped) that may be sensitive to underlying ground truth image structures and properties. Showing some results for common datasets (with complicated textures and patterns) as well as some synthetic noise with different noise levels (ideal cases) will convince readers that the proposed method can be generalized to other cases. The proposed method involves noise estimation and it seems to work well with texts, faces and microscopy imaging. However, these results do not seem to be easily extended to other imaging cases.

In addition, it seems that generating 1000 images for averaging will be computationally expensive. Thus, in Table 1, it will be fair to indicate computation time for training / testing per image.

---

> ### Author Response · Authors · 2020-11-11
> **Initial Point by Point Rebuttal (and invitation for further discussion)**
>
> We very much appreciate the insightful comments in this review.
> Here, we will comment on all points while maintaining the proposed order and numbering scheme of the review.
>
> 1. **Comparison to other methods:**
> Quan et al. (Self2Self) describe an interesting denoising method, but it is very different to our method (DivNoising). Self2Self is not based on VAEs but on a U-Net-like architecture with blind-spots and dropout. Hence, it leverages ‘model uncertainty’, while we are the first to approximate the real (data) posterior, allowing us to get a grip on the ‘data uncertainty’.
> We stress that unlike Self2Self, DivNoising uses an explicit noise model $p(x|s)$.
> **PROPOSAL**: we discuss Self2Self in our related work section.
>
> **a. Training on single inputs vs. whole datasets:**
> Self2Self is trained on single images. For 4 randomly chosen single images from 4 datasets, we have now run Self2Self and compared it to DivNoising (i) trained on same single image, and (ii) trained on entire dataset. DivNoising leads to superior results in a fraction of the training time. Applying Self2Self to a full dataset would take an impractical amount of time (about 1000 GPU hours vs. 10 GPU hours for DivNoising). Biomedical research datasets typically consist of many images (GB or TB).
> **PROPOSAL**: we include Self2Self as additional baseline in appendix and refer to it in main text.
>
> **Comparison of Self2Self (left) to DivNoising trained on same single image (middle) and DivNoising trained on full dataset (right) :**
>
> PSNR (dB)
> * Convallaria: 36.23; 36.42; **36.94**
> * Actin: 33.15; 33.80; **33.99**
> * Nuclei: 36.21; 35.99; **36.46**
> * W2S Ch1 av1: 31.40; **31.81**; 31.59
>
> Training time (hours)
> * Convallaria: 10; **0.44**; 10
> * Actin: 10; **1.09**; 10
> * Nuclei: 10; **0.16**; 7
> * W2S Ch1 av1: 10; **0.48**; 3.75
>
> Required GPU memory (GB)
> * Convallaria: 11; **1.5**; **1.5**
> * Actin: 11; **1.5**; **1.5**
> * Nuclei: 11; **1.5**; **1.5**
> * W2S Ch1 av1: 11; **1.5**; **1.5**
>
> **b. Baselines and benchmark datasets used:**
> We propose a method for denoising of biomedical microscopy data, hence we are evaluating our performance on benchmark datasets widely used in this domain (see cited literature in paper) and compare to state-of-the-art baselines.
> The official call for papers for ICLR 2021 explicitly lists applications in computational biology as a subject area (https://iclr.cc/Conferences/2021/CallForPapers), which initially motivated us to submit our work to ICLR.
> All this being said, DivNoising is by no means limited to biomedical images. To this end, we show that commonly used applications such as OCR and face recognition in forensics can also benefit from our ideas. (Results on natural images BSD68 dataset are already in Appendix Fig. 25.)
> **PROPOSAL**: we make the scope of the paper clearer in introduction and include a paragraph on the importance of denoising for computational biology.
>
> **c. Single channel vs. RGB data:**
> Microscopy images can have multiple fluorescence channels, but those are never RGB channels and are typically processed independently to avoid undesired crosstalk. Contrary to reviewer's remark, we show results on both real and synthetic data.
> **PROPOSAL**: we include additional baselines or single-channel (microscopy related) datasets if reviewers think it is strictly required.
>
> **Sampling scheme already proposed in [Kohl2018]?**
> The paper by Kohl et al. is great, but by no means the first to propose the sampling scheme we use. Initially introduced by Kingma et al. (2014), it has been used by many papers and we discussed this in our Related Work section. Our contribution is the way we model the posterior of VAEs. We ask the reviewer to revisit our related work, where we have also explicitly mentioned and discussed [Kohl2018].
>
> 2. **DivNoising on other image domains + computation time**
> Extending (generalizing) DivNoising to other image domains is an interesting future work. Here, we show how we can deal with dominant noises in microscopy: Poisson shot noise and Gaussian readout noise (and also salt&pepper noise). Hence, DivNoising will help many biomedical researchers analyze their data! Going beyond this (within biomedical domain) will extend to e.g. MRI or CT (tomography) data, but with the right noise model, we believe many domains can benefit from our general ideas.
> **PROPOSAL**: we extend the paragraph in the discussion section that points at challenges ahead.
>
> **Runtime for sampling 1000 DivNoised images (e.g. for MMSE estimation)**
> This sampling, only needed during prediction, requires less than 7 seconds per image even for the largest image in all our datasets.
> **PROPOSAL**: we will include precise sampling time in final paper.
>
> **Finally,** we want to encourage the reviewer to comment on our proposals in order to let us understand if these are adequate measures to address her/his concerns. If we misunderstood some of the voiced criticisms, please critique us again as soon as possible.
> _Thanks!_ :)

---

### Author Response · Authors · 2020-11-17
**Revised Paper Uploaded**

We thank everyone for their constructive feedback. We are happy to see that our paper was generally well received by all reviewers.

We uploaded a modified PDF containing changes suggested by the reviewers' comments and corresponding to our proposals made in the rebuttal.

Please note that for easy reviewing, all changes are marked in blue.

Additionally, we wish to clarify that a remark by Reviewer 3 about the limited resolution of input images in our datasets may have been a result of misunderstanding. Some of our benchmark biomedical datasets are already 1024x1024 and our method works very well at denoising those images. We have explained this in detail in our response below to Reviewer 3's comments. In short, our method is not restricted by the resolution of input images.

If the reviewers have additional remarks, we are happy to engage in further discussion.

Thanks,

the authors

---

### Author Response · Authors · 2021-04-01
**Code Released Publicly**

We have released our code publicly at https://github.com/juglab/DivNoising.

---

### Decision · Program_Chairs · 2021-01-07
**Final Decision**

**Decision:**

Accept (Poster)

**Comment:**

A simple but sensible idea to improve VAE with good experimental results.